# γδ T cells control humoral immune response by inducing T follicular helper cell differentiation

Rafael M. Rezende[1], Amanda J. Lanser[1], Stephen Rubino[1], Chantal Kuhn[1], Nathaniel Skillin[1], Thais G. Moreira[1], Shirong Liu[1], Galina Gabriely[1], Bruna A. David[2], Gustavo B. Menezes[2] & Howard L. Weiner[1]

γδ T cells have many known functions, including the regulation of antibody responses. However, how γδ T cells control humoral immunity remains elusive. Here we show that complete Freund's adjuvant (CFA), but not alum, immunization induces a subpopulation of CXCR5-expressing γδ T cells in the draining lymph nodes. TCRγδ$^+$CXCR5$^+$ cells present antigens to, and induce CXCR5 on, CD4 T cells by releasing Wnt ligands to initiate the T follicular helper (Tfh) cell program. Accordingly, TCRδ$^{-/-}$ mice have impaired germinal center formation, inefficient Tfh cell differentiation, and reduced serum levels of chicken ovalbumin (OVA)-specific antibodies after CFA/OVA immunization. In a mouse model of lupus, TCRδ$^{-/-}$ mice develop milder glomerulonephritis, consistent with decreased serum levels of lupus-related autoantibodies, when compared with wild type mice. Thus, modulation of the γδ T cell-dependent humoral immune response may provide a novel therapy approach for the treatment of antibody-mediated autoimmunity.

[1] Ann Romney Center for Neurologic Diseases, Brigham and Women's Hospital, Harvard Medical School, Boston, MA 02115, USA. [2] Center for Gastrointestinal Biology, Federal University of Minas Gerais, Belo Horizonte, Minas Gerais 31270-901, Brazil. Correspondence and requests for materials should be addressed to R.M.R. (email: rmachadorezende@bwh.harvard.edu)

Antibody production is a multi-step process involving CD4[+] T cell activation, their differentiation into T follicular helper (Tfh) cells, germinal center formation, immunoglobulin class switching (also known as isotype switching), affinity maturation, plasma cell development, and memory B cell generation[1,2].

Naïve CD4[+] T cells differentiate into Tfh cells in response to IL-6, inducible costimulator (ICOS), and T cell receptor (TCR) signaling[3–6]. Recently, the transcription factor achaete-scute homologue-2 (Ascl2) was shown to initiate the Tfh development[7]. In a mechanism involving the β-catenin pathway, naïve CD4[+] T cells upregulate Ascl2, thus initiating the Tfh program that involves CXCR5 upregulation, CCR7 downregulation, and Th1 and Th17 gene signature inhibition[7]. However, the source of endogenous β-catenin activation molecules (Wnt agonists) is not known. The Tfh cell program is then maintained by expression of transcription factor B cell lymphoma 6 (Bcl6)[1].

Once differentiated, Tfh cells migrate to the B:T cell border of a lymphoid organ, where they encounter cognate antigen-activated B cells. This Tfh–B cell interaction results in B cell proliferation and differentiation. B cells then migrate to the center of the follicle and give rise to the germinal center where isotype switching and antibody affinity maturation take place[2].

In the absence of αβ T cells, B cells are able to expand and secrete copious amounts of T cell-dependent antibodies, which react to self-antigens, mimicking the pathogenesis of systemic lupus erythematosus (SLE)[8]. Thus, "non-αβ" T cells can mediate immunoglobulin class switching and antigen-dependent antibody production, suggesting that γδ T cells play an important role in these processes. In fact, it has been shown that γδ T cell deficient (TCRδ[−/−]) mice, either immunized or not, have reduced serum antibody levels, including IgG1, IgG2b, and IgE[9,10]. Importantly, some of these antibody subclasses, such as IgG2b and IgG2c were αβ T cell independent whereas IgG1 and IgE were αβ T cell dependent. Interestingly, the hypogammaglobulinemia observed in TCRδ[−/−] mice depends on the specific γ gene deletion. For example, Vγ1 knockout mice have hypogammaglobulinemia, whereas Vγ4 and Vγ6 double-knockout mice have increased serum antibody levels, particularly IgE, compared to wild-type (WT) mice, an effect likely to be dependent on IL-4[10]. This suggests that γδ T cell-dependent antibody production involves both αβ T cell dependent and independent pathways and that this effect is controlled by the cross-talk between γδ T cell subsets.

In humans, γδ T cells promote B cell somatic hypermutation and isotype switching by expressing several factors: (1) CXCR5[11], a chemokine receptor that allows migration toward CXCL13 in the B cell follicle; (2) CD40 ligand (CD40L)[12], crucial for B cell activation, and (3) IL-4 and IL-10 cytokine secretion[11], involved in immunoglobulin class switch. Consistent with this, γδ T cells have been implicated in antibody-mediated autoimmune diseases such as SLE. Notably, pathogenic anti-DNA autoantibody-inducing γδ T cell lines were isolated from patients with active lupus nephritis[13]. Moreover, a subgroup of patients with SLE and Sjogren's syndrome displayed a marked increase in γδ T cell numbers that were normalized by immunosuppressant treatment[14]. Thus, these studies suggest the involvement of γδ T cells in antibody-mediated autoimmune conditions. However, the mechanisms underlying γδ T cell-dependent humoral immunity remain elusive. For example, whether γδ "Tfh-like" cells exist or whether γδ T cells communicate directly with B cells or interfere with Tfh cell development.

Here, we show that upon immunization with CFA, but not Alum, CXCR5 expression is induced on γδ T cells in a γδTCR activation-dependent fashion. TCRγδ[+]CXCR5[+] cells secrete Wnt ligands that induce CXCR5 expression on CD4[+] T cells, leading to their differentiation into Tfh cells. Consistent with this,

TCRδ[−/−] mice show reduced Tfh cell frequencies and germinal center formation and have decreased production of both OVA-specific antibodies and self-reactive antibodies compared to WT mice. Moreover, in a murine model of lupus, TCRδ[−/−] mice develop milder glomerulonephritis compared to WT mice. These data advance our understanding of the mechanisms by which γδ T cells control humoral immune responses and promote antibody-mediated diseases.

## Results

**Reduced anti-OVA antibodies in CFA-immunized TCRδ[−/−] mice.** To investigate whether γδ T cell affected specific antibody production and whether αβ T cells were required for the ability of γδ T cells to control humoral immunity, WT, TCRδ[−/−], and TCRβ[−/−] mice were immunized with either CFA/OVA or Alum/OVA at day 0, received an OVA boost at day 14 and sera and feces (for secretory (s)IgA analysis) for OVA-specific antibody measurement were collected at day 21. We found that, upon CFA immunization, serum levels of OVA-specific immunoglobulin levels were reduced in TCRδ[−/−] mice (Fig. 1a–h). Importantly, none of the antibodies measured in TCRδ[−/−] mice were decreased after Alum immunization (Fig. 1a–h), suggesting that the γδ T cell control of specific antibody production depends on the adjuvant used for immunization. Furthermore, in the absence of αβ T cells, γδ T cells did not restore the diminished OVA-specific antibody in the sera and feces of TCRβ[−/−] mice regardless of the type of adjuvant used for immunization (Fig. 1a–h), suggesting that the mechanism by which γδ T cells regulate humoral immune response is αβ T cell dependent.

It has been shown that lower antibody levels in TCRδ[−/−] mice relates to the absence of the Vγ1 subset, which favors humoral immune responses, as opposed to Vγ4 subset that suppresses humoral responses[9,10,15,16]. Thus, the balance between Vγ1 and Vγ4 γδ T cell subset functions determines the level of antibody production following CFA vs. Alum immunization and could explain why γδ T cells do not participate in antibody production in Alum immunized mice (Fig. 1). To test this, we depleted Vγ4 γδ T cells with an anti-Vγ4-depleting mAb and found marked increase in all OVA-specific immunoglobulin antibodies, except for sIgA, following CFA immunization (Supplementary Fig. 1a–h). Alum immunization however only increased IgM levels in Vγ4 γδ T cell-depleted mice (Supplementary Fig. 1b). Thus, even in the absence of the inhibitory effects of Vγ4 γδ T cells, Vγ1 γδ T cells do not promote antibody production following Alum immunization and an imbalance in Vγ1/Vγ4 γδ T cell function is unlikely to be responsible for the differential involvement of γδ T cells in antibody production following CFA vs. Alum immunization.

**Reduced autoantibodies in TCRδ[−/−] mice.** It has been shown that in the absence of αβ T cells, B cells are able to expand and secrete T cell-dependent antibodies, which react to self-antigens[8]. To test whether TCRδ[−/−] mice have decreased autoantibody production, we performed an autoantibody microarray on serum from naïve WT vs. naïve TCRδ[−/−] mice. We found that several self-reactive IgM and IgG antibodies were diminished in TCRδ[−/−] mice compared to WT mice (Fig. 2a, b; Supplementary Data 1). IgM autoantibodies that were decreased in TCRδ[−/−] mice included those reactive to nuclear components (histones, nucleosome antigen, SmD1, U1-snRNP-A, Sm/RNP), complement factors, fibrinogen S, aquaporin 4 (AQP4), myelin-associated glycoprotein (MAG), and muscarinic receptor (Fig. 2a; Supplementary Data 1). Similarly, IgG autoantibodies to nuclear components (ssDNA, histones, U1-snRNP-A), complement factors, C-reactive protein (CRP), fibrinogen S, and AQP4

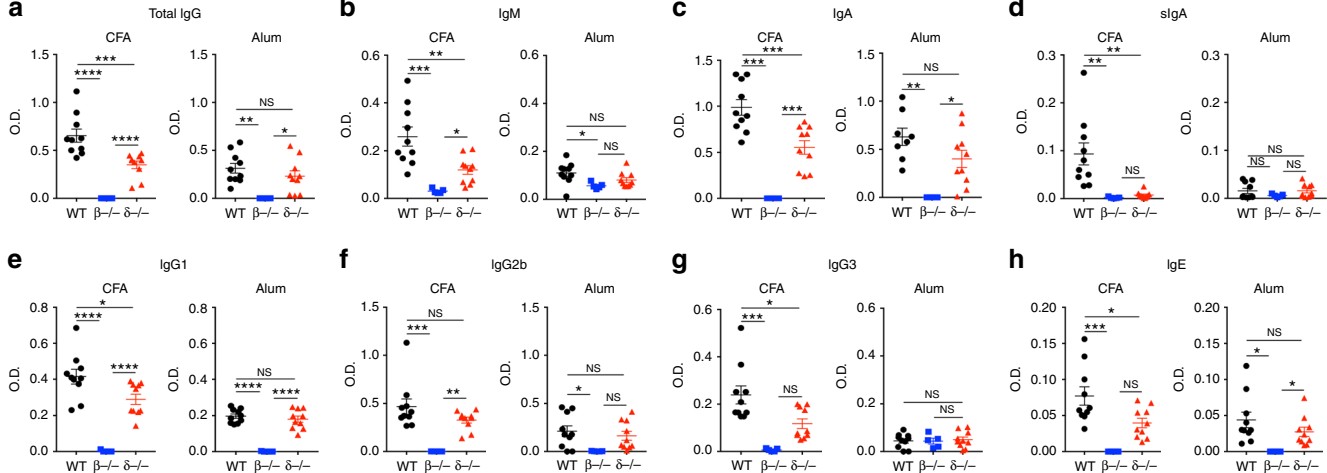

**Fig. 1** OVA-specific antibodies in TCRδ$^{-/-}$ mice. **a–h** Serum and fecal (secretory IgA) OVA-specific total IgG (**a**), IgM (**b**), IgA (**c**), secretory IgA (sIgA; **d**), IgG1 (**e**), IgG2b (**f**), IgG3 (**g**), and IgE (**h**) from WT, TCRβ$^{-/-}$, and TCRδ$^{-/-}$ mice 21 days after s.c. immunization with either OVA + CFA or OVA + Alum. Mice received a boost of OVA 14 days after immunization (n = 5–10 mice/group). These data are representative of three independent experiments. Data are shown as mean + SEM. One-way ANOVA was used. NS non-significant, *p < 0.05, **p < 0.01, ***p < 0.001, ****p < 0.0001

were decreased in TCRδ$^{-/-}$ mice (Fig. 2b; Supplementary Data 1).

Because increased levels of serum antibodies reactive to nuclear components and complement factors are known to be involved in the pathogenesis of SLE[17], we investigated whether TCRδ$^{-/-}$ mice are protected from pristane-induced lupus. Pristane is an isoprenoid alkane found at high concentration in mineral oil that is known to induce autoantibodies characteristic of lupus in mice[18]. We found increased levels of both IgM and IgG autoantibodies 3 months after pristane injection in WT mice (Supplementary Data 1). IgG self-reactive antibodies, including anti-nuclear and anti-complement antibodies, were decreased in TCRδ$^{-/-}$ mice (Fig. 2c; Supplementary Data 1). Excess autoantibodies deposit in the kidneys and induce glomerulonephritis characterized by enlarged and hypercellular glomeruli after pristane treatment[18,19]. Five months after pristane injection, we found that TCRδ$^{-/-}$ mice had milder signs of glomerulonephritis compared to WT mice (Fig. 2d). Thus, TCRδ$^{-/-}$ mice had reduced self-reactive antibody levels and were protected from kidney damage caused by antibody deposition.

**Tfh cell induction is impaired in TCRδ$^{-/-}$ mice.** The impaired antibody production in TCRδ$^{-/-}$ mice could be related to dysregulated B cells and/or abnormalities in Tfh cells. To investigate these possibilities, we immunized mice with CFA to induce Tfh cells[20,21] and investigated both peripheral and bone marrow B cell compartments 7 days after immunization. As expected[22], no major changes between WT and TCRδ$^{-/-}$ mice in both B cell compartments were observed (Supplementary Fig. 2a-c), despite decreased frequencies of activated B cells in the draining lymph node (dLN) of TCRδ$^{-/-}$ mice (Supplementary Fig. 2b), suggesting the involvement of γδ T cells in B cell activation after antigen exposure. In the bone marrow, we did not observe major differences between WT and TCRδ$^{-/-}$ mice and between immunized and non-immunized mice apart from a slight increase in pre-pro B cell frequency and a decrease in long-lived plasma cells 7 days post-immunization (Supplementary Fig. 2c). Thus, the impaired humoral immune response in TCRδ$^{-/-}$ mice is not due to a defect in the B cell compartment.

We next investigated whether γδ T cells are involved in Tfh cell differentiation and/or function. For this, we immunized WT and TCRδ$^{-/-}$ mice and examined the dLN architecture and cell composition 1, 3, 5, and 7 days after immunization. We imaged LNs ex-vivo by employing a method we developed that improves the speed, structure preservation, and cell viability compared to conventional immunofluorescence staining (Supplementary Fig. 3a). We found that dLNs from WT mice had a normal structure with well-defined follicles and a T cell zone as compared to TCRδ$^{-/-}$ mice (Fig. 3a). Moreover, in WT mice, γδ T cells were present in the follicles even in non-immunized mice, and γδ T cell number increased from day 3 after immunization, where they accumulated in the T cell zone and within the follicles (Fig. 3a). Thus, γδ T cells may play an important role in maintaining the homeostasis of lymphoid organs.

To investigate the ability of TCRδ$^{-/-}$ mice to form germinal centers (GC), we measured B220$^+$FAS$^+$GL7$^+$ expression in WT and TCRδ$^{-/-}$ mice 7 days after immunization. The frequencies of B220$^+$FAS$^+$GL7$^+$ cells in TCRδ$^{-/-}$ mice were comparable to those in non-immunized mice and significantly lower than in immunized WT mice (Fig. 3b). We confirmed the reduced GC formation in TCRδ$^{-/-}$ mice by whole lymph node confocal microscopy (Supplementary Fig. 3b). Thus, TCRδ$^{-/-}$ mice have impaired GC formation after immunization.

We then investigated whether TCRδ$^{-/-}$ cells mice had a defect in the Tfh cell compartment, which could explain their impaired GC formation. We analyzed Tfh cells in the dLNs, non-draining lymph nodes (ndLNs) and spleen of WT vs. TCRδ$^{-/-}$ mice 1, 3, 5, 7, and 21 days after immunization. In dLNs, we found that TCRδ$^{-/-}$ mice had 50% less CXCR5$^+$ expression on CD4$^+$ T cells 1, 3, 5 and 7 days post-immunization (Fig. 3c). Consistent with this, CD4$^+$CXCR5$^+$Bcl6$^+$ Tfh cells were 50% less frequent in TCRδ$^{-/-}$ vs. WT mice 7 days post-immunization (Fig. 3d). Of note, Tfh cell frequencies returned to basal levels 21 days after immunization and no difference between WT and TCRδ$^{-/-}$ was observed at this time (Supplementary Fig. 3c). There was no difference in Tfh cell frequencies in ndLNs (Supplementary Fig. 4a) or spleen (Supplementary Fig. 4b) between non-immunized and immunized mice. The decrease in Tfh cells in dLNs was confirmed by confocal microscopy, where minimal CXCR5 expression was detected in TCRδ$^{-/-}$ mice (Fig. 3e). Interestingly, we also observed CXCR5 staining in close proximity to γδ-GFP cells by confocal microscopy (Fig. 3e, arrows), suggesting that a subtype of CXCR5-expressing γδ T cells may exist. Importantly, in the pristane-induced lupus model, the frequency of Tfh cells was decreased and correlated with lower

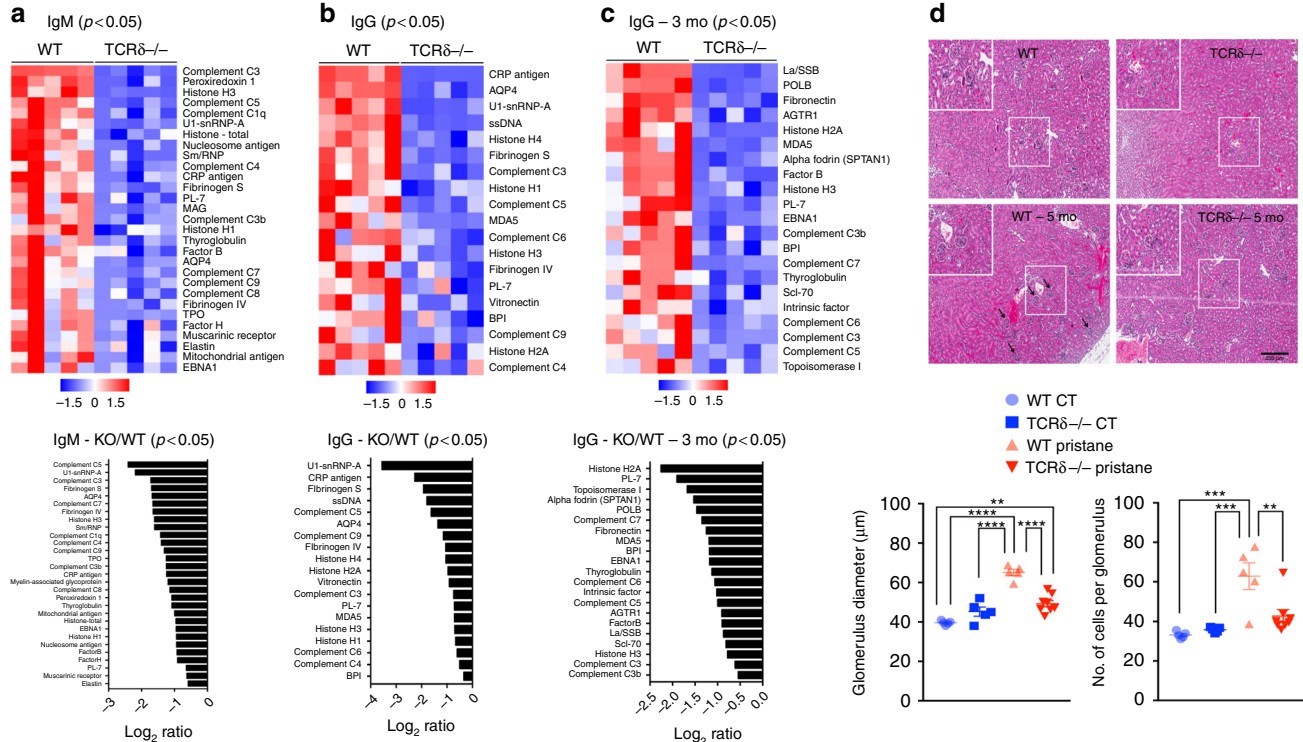

**Fig. 2** TCRδ$^{-/-}$ mice are less affected by pristane-induced lupus. **a–c** Heat map and log$_2$ ratio of serum levels ($p < 0.05$) of IgM (**a**), IgG (**b**) autoantibodies from naïve TCRδ$^{-/-}$ vs. WT mice, and IgG autoantibodies (**c**) 3 months after pristane injection ($n = 5$ mice/group). An autoantigen microarray super panel (128 autoantigens) were used to generate data. The averaged net fluorescent intensity (NFI) of each autoantigen was normalized to internal controls (IgG or IgM; see Methods). Statistic different autoantigens ($p < 0.05$, WT vs. TCRδ$^{-/-}$ mice) were used to plot data. **d** Representative histologic sections and quantification of glomerulus diameter and number of cells per glomerulus of kidneys from WT and TCRδ$^{-/-}$ mice removed 5 months after pristane injection with 5 μm serial sections stained with hematoxylin–eosin. Magnification of 50×. Scale bar = 250 μm ($n = 5$–8 mice/group). Data are shown as mean + SEM. Student's $t$-test (**a–c**) or One-way ANOVA (**d**) were used. *$p < 0.05$, **$p < 0.01$, ***$p < 0.001$, ****$p < 0.0001$

sera autoantibodies in TCRδ$^{-/-}$ mice 5 months after pristane injection as compared to WT mice (Supplementary Fig. 4c).

TCRδ$^{-/-}$ mice immunized with CFA/OVA also had 50% less OVA-specific Tfh cells, as measured by the expression of the OVA-specific TCR Vβ5.1/5.2 on CD4$^+$CXCR5$^+$Bcl6$^+$ cells (Supplementary Fig. 5a). Furthermore, because we found that TCRδ$^{-/-}$ mice upon Alum immunization had normal levels of antibodies (Fig. 1), we immunized mice with either CFA or Alum and found that, despite a higher frequency of Tfh cells in CFA-immunized mice, no differences between WT and TCRδ$^{-/-}$ mice were found after Alum immunization (Supplementary Fig. 5b). Moreover, Foxp3-expressing T follicular regulatory (Tfr) frequency, which has been shown to play an important role in GC control[23], was not different between WT and TCRδ$^{-/-}$ mice immunized with CFA, but was decreased compared to non-immunized mice (Supplementary Fig. 5c).

Importantly, Tfh cells from TCRδ$^{-/-}$ mice expressed less CD40L than Tfh cells from WT mice after immunization, although PD-1, ICOS, and IL-21 had similar expression levels (Supplementary Fig. 6a-d). This suggests that Tfh cells from TCRδ$^{-/-}$ mice do not provide adequate help to B cells for proper GC formation and subsequent antibody production.

Taken together, these data demonstrate that γδ T cells are important for CXCR5 induction on CD4$^+$ T cells and for the ability of Tfh cells to provide help for B cells. Disruption of this interaction is likely the cause of the impaired humoral immune response in TCRδ$^{-/-}$ mice.

**A subtype of γδ T cell expresses CXCR5.** Because we found γδ T cells within the follicles, and close proximity of CXCR5 staining

to γδ-GFP (Fig. 3a, e), we asked whether γδ T cells express CXCR5. We examined CXCR5 protein expression on γδ T cells in non-immunized and 1, 3, 5, and 7 day-immunized mice. Different from CD4$^+$ T cells that showed peak of CXCR5 expression at day 7 post-immunization, the peak of CXCR5 expression on γδ T cells was observed 1 day after immunization and remained constant up to day 7 (Fig. 4a). Importantly, CXCR5 expression was only upregulated on Vγ1 γδ T cells after CFA immunization (Fig. 4b), which is consistent with the important role of Vγ1 γδ T cells in antibody production control (Supplementary Fig. 1). Furthermore, γδ T cells from mice with pristane-induced lupus also had increased CXCR5 expression compared to naïve mice (Supplementary Fig. 7a). However, CXCR5 expression on total γδ T cells (Supplementary Fig. 7b) or Vγ1 γδ T cells (Supplementary Fig. 9a) was not upregulated after Alum immunization as compared to mice immunized with CFA. Consistent with the increased levels of OVA-specific immunoglobulins following Vγ4 γδ T cell depletion, we found that Tfh cell frequency was higher in Vγ4 γδ T cell-depleted mice than in isotype control (IC)-treated mice immunized with CFA/OVA, but not after Alum/OVA immunization (Supplementary Fig. 7c). Importantly, TCRδ$^{-/-}$ mice transferred with CXCR5$^{-/-}$ γδ T cells had decreased Tfh cell frequencies and did not develop GC following CFA immunization as compared to mice transferred with WT γδ T cells (Fig. 4c, d), confirming that CXCR5 expression on γδ T cells is crucial for their ability to induce Tfh cells and to control the humoral immune response.

To investigate whether γδ T cells shared Tfh cell features, we measured Bcl6 expression in γδ T cells in non-immunized and 1, 3, 5, and 7 day-immunized mice and found no Bcl6 expression

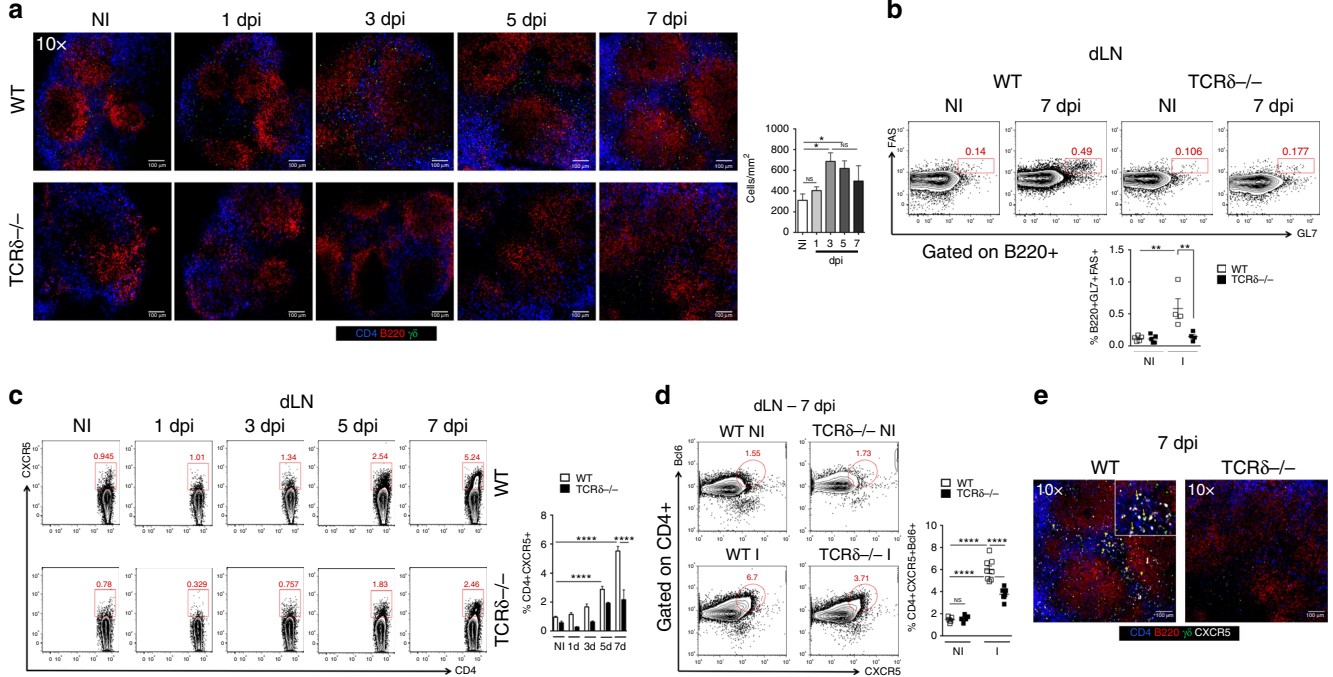

**Fig. 3** TCRδ−/− mice have reduced Tfh cell and GC compartments. **a** Representative confocal microscopy images of whole lymph nodes (dLN) of non-immunized (NI) TCRγδ-GFP and TCRδ−/− mice and 1, 3, 5, and 7 days post CFA immunization (dpi). Scale bar = 100 μm. CD4—blue; B220—red; TCRγδ—green. Bar graph represents the number of GFP+ cells per mm2 (n = 6 mice/group). **b** Frequency of B220+ cells expressing FAS and GL7 (indicative of germinal center cells) in the dLNs of non-immunized (NI) WT and TCRδ−/− mice and 7 days post CFA immunization (dpi) (n = 4 mice/group). **c** Frequency of CD4+ T cells expressing CXCR5 in the dLNs of non-immunized (NI) WT and TCRδ−/− mice and 1, 3, 5, and 7 days post CFA immunization (dpi) (n = 6 mice/group). **d** Frequency of CD4+ T cells expressing CXCR5 and Bcl6 in dLNs of non-immunized (NI) WT and TCRδ−/− mice and 7 days after CFA immunization (I) (n = 5–8 mice/group). **e** Representative confocal microscopy images of whole lymph nodes (dLN) of non-immunized (NI) TCRγδ-GFP and TCRδ−/− mice, and 7 days post CFA immunization (dpi). Scale bar = 100 μm. CD4—blue; B220—red; TCRγδ—green; CXCR5—gray. Yellow arrows indicate γδ T cells and CXCR5 co-localization (n = 6 mice/group). These data are representative of three independent experiments. Data are shown as mean + SEM. One-way ANOVA was used. *p < 0.05, **p < 0.01, ****p < 0.0001

(Fig. 4e). Moreover, no difference between TCRγδ+CXCR5+ and TCRγδ+CXCR5− cells was observed for PD-1, ICOS, IL-21R, IFN-γ, and IL-17A (Supplementary Fig. 7d-f). Neither TCRγδ+CXCR5+ nor TCRγδ+CXCR5− cells produced IL-21 or expressed ICOSL (Supplementary Fig. 8a). Both TCRγδ+CXCR5+ and TCRγδ+CXCR5− cells expressed low levels of CD40L (Supplementary Fig. 8a). Importantly, even in the absence of αβ T cells (TCRβ−/− mice), TCRγδ+CXCR5+ cells did not upregulate CD40L nor IL-21 (Supplementary Fig. 8b), suggesting that γδ T cells do not provide direct help to B cells in a cell–cell contact fashion.

Thus, TCRγδ+CXCR5+ cells represent a subtype that does not share common features with Tfh cells and are not a γδ Tfh cell subset.

**γδTCR activation induces CXCR5 expression on γδ T cells.** Because CFA, but not Alum, induced CXCR5 expression on γδ T cells (Fig. 5a; Supplementary Fig. 7b), we chose the stimuli that could be involved in CFA stimulation to investigate potential mechanisms by which CXCR5 is induced on γδ T cells. We found that γδTCR, but not TLR1/2 stimulation, found to be highly expressed on both TCRγδ+CXCR5+ and TCRγδ+CXCR5− cells (Supplementary Data 2), induced CXCR5 on γδ T cells. Addition of either anti-CD28 or TLR1/2 agonist in the culture did not potentiate the CXCR5 expression induced by anti-TCRγδ. Moreover, CXCR5 expression on γδ T cells was also upregulated by Mycobacterium tuberculosis (MT) and this effect was neither blocked by the TLR1/2 antagonist CU-CPT22 nor potentiated by the addition of anti-TCRγδ (Fig. 5b), suggesting that CXCR5

expression on γδ T cells is induced by γδTCR activation. Consistent with this, in vivo administration of anti-TCRγδ monoclonal antibody upregulated CXCR5 on γδ T cells from both spleen and inguinal lymph node (ILN) 24 h after injection (Fig. 5c). Thus, CXCR5 expression on γδ T cells is induced by γδTCR activation.

**TCRγδ+CXCR5+ cells function as antigen-presenting cells.** To further characterize TCRγδ+CXCR5+ cells, we sorted TCRγδ+CXCR5+ and TCRγδ+CXCR5− cells and measured gene expression using the Nanostring nCounter mouse immunology codeset. Hierarchical clustering showed that TCRγδ+CXCR5+ and TCRγδ+CXCR5− cells are distinct populations (Fig. 6a). Twenty genes were decreased in the TCRγδ+CXCR5+ vs. TCRγδ+CXCR5− cells (with p < 0.05; Fig. 6a, b; Supplementary Data 2). In contrast, 65 genes were increased in TCRγδ+CXCR5+ vs. TCRγδ+CXCR5− cells with (p < 0.05). Among them were those related to antigen presentation, such as MHC class II molecules (H2-Aa, H2-Ab1, H2-Eb1), CD40 and CD86 (Fig. 5a, b; Supplementary Data 2). We confirmed the protein expression of MHC-II, CD40, and CD86 on TCRγδ+CXCR5+ vs. TCRγδ+CXCR5− cells by flow cytometry (Fig. 5c). Consistent with the role of Vγ1+ γδ T cells in controlling antibody production (Supplementary Fig. 1), high levels of MHC-II were upregulated exclusively on Vγ1+ γδ T cell following CFA, but not following Alum immunization. Moreover, CXCR5 expression was only observed on Vγ1+ MHC-IIhigh γδ T cells (Supplementary Fig. 9a).

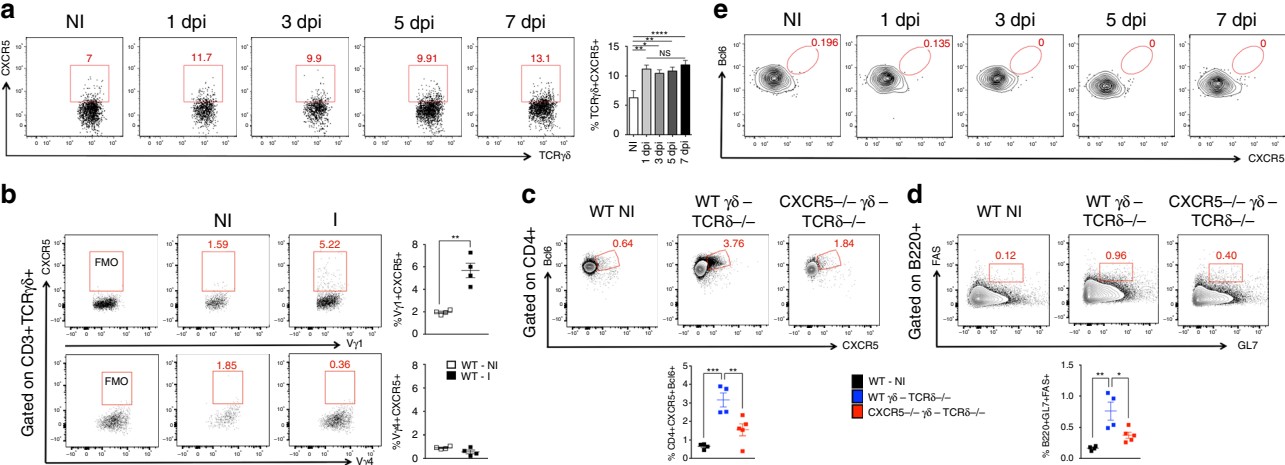

**Fig. 4** A subset of γδ T cells expresses CXCR5. **a** Frequency of γδ T cells expressing CXCR5 in the dLN of non-immunized (NI) WT mice and 1, 3, 5, and 7 days post CFA immunization (dpi) ($n = 6$ mice/group). **b** Frequency of Vγ1 and Vγ4 γδ T cells expressing CXCR5 from dLN of non-immunized (NI) WT mice and 3 days after CFA immunization (I) ($n = 4$ mice/group). **c**, **d** In vivo Tfh cell induction (**c**) and germinal center formation (**d**) in dLNs of TCRδ$^{-/-}$ mice transferred intravenously with $5 \times 10^5$ γδ T cells from either WT or CXCR5$^{-/-}$ mice. Three days after transfer, recipient mice were immunized or not (WT NI) with CFA and 7 days thereafter sacrificed for flow cytometric analysis ($n = 4$–5 mice/group, where γδ T cells are sorted from a pool of 10 mice/group). **e** Representative FACS plots showing frequency of γδ T cells expressing CXCR5 and Bcl6 in dLN of non-immunized (NI) WT and 1, 3, 5, and 7 days post CFA immunization (dpi) ($n = 5$ mice/group/time point). These data are representative of at least two independent experiments (**a**, **b**, **e**). Data are shown as mean + SEM. One-way ANOVA was used. NS non-significant, $*p < 0.05$, $**p < 0.01$, $***p < 0.001$, $****p < 0.0001$

To determine whether TCRγδ$^+$CXCR5$^+$ cells functioned as APCs, we cultured TCRγδ$^+$CXCR5$^+$, TCRγδ$^+$CXCR5$^-$, or CD11c$^+$ dendritic cells (DC) with Alexa Fluor 488-conjugated ovalbumin (OVA). TCRγδ$^+$CXCR5$^+$ cells took up OVA, but to a lesser extent than CD11c$^+$ DCs (21.5% vs. 49.8%, respectively). No OVA uptake was observed in TCRγδ$^+$CXCR5$^-$ cells (Fig. 5d). When OVA$_{323-339}$ peptide-pulsed TCRγδ$^+$CXCR5$^+$ cells were cultured with naïve CD4$^+$ T cells from OT-IIxFoxp3-GFP (OVA$_{323-339}$-specific TCR transgenic) mice, responder T cells proliferated in the presence of TCRγδ$^+$CXCR5$^+$ cells and to a greater extent in the presence of CD11c$^+$ DCs (52.4% and 98%, respectively). However, consistent with results from the uptake assay, no proliferation was observed when responder T cells were co-cultured with TCRγδ$^+$CXCR5$^-$ cells, when TCRγδ$^+$CXCR5$^+$ cells or CD11c$^+$ DCs were not pulsed with OVA$_{323-339}$ peptide or when TCRγδ$^+$CXCR5$^+$ and CD11c$^+$ DCs were sorted from MHC-II$^{-/-}$ mice (Fig. 5e). Thus, these results demonstrate that CXCR5-expressing γδ T cells possess antigen presentation properties. Confirming the importance of antigen presentation ability of γδ T cells to induce Tfh cell differentiation and GC formation in vivo, TCRδ$^{-/-}$ mice transferred with MHC-II$^{-/-}$ γδ T cells had decreased Tfh cell frequencies and did not develop GC following CFA immunization as compared to mice transferred with WT γδ T cells (Fig. 6f, g).

**TCRγδ$^+$CXCR5$^+$ cell-secreted Wnt ligands initiate the Tfh cell program**. It has been recently shown that Wnt ligands initiate the Tfh cell program through the upregulation of the transcription factor Ascl2 in naïve CD4$^+$ T cells[7]. Because Tfh cell differentiation takes place within a secondary lymphoid organ after antigen exposure, some cell in the T cell zone must release endogenous Wnt ligands. Two possible candidates are DCs and TCRγδ$^+$CXCR5 cells, since antigen presentation and CD4$^+$ T cell activation is the first step in Tfh cell differentiation. To investigate these possibilities, we first measured Ascl2 mRNA expression in sorted CD4$^+$CD44$^+$CD25$^-$ T cells from dLNs of WT or TCRδ$^{-/-}$ mice before immunization and 1, 2, 3, and 4 days after immunization. We found that immunized WT mice upregulated Ascl2, which peaked at day 3 and decreased at day 4 post-immunization

(Fig. 7a). These findings are consistent with the CXCR5 expression kinetics we observed on CD4$^+$ T cells (Fig. 3c). CD4$^+$CD44$^+$CD25$^-$ T cells from TCRδ$^{-/-}$ mice minimally upregulated Ascl2 (Fig. 7a), consistent with the lower expression of CXCR5 on CD4$^+$ T cells in these mice (Fig. 3c).

To determine the source of endogenous Wnt ligands in the lymph node, we sorted γδ T cells and CD11c$^+$ DCs from dLN of WT mice before immunization and 1, 2, 3, and 4 days after immunization and measured the mRNA expression of several Wnt ligands involved in the canonical (β-catenin-dependent), including Wnt1, Wnt2, Wnt3a, Wnt4, Wnt6, Wnt7a, Wnt7b, Wnt8a, Wnt8b; and non-canonical (β-catenin independent) pathways, such as Wnt5a and Wnt11. We found that Wnt6 was highly expressed in CD11c$^+$ DCs from non-immunized mice, but was downregulated from day 1 to 3 after immunization and returned to the basal level at day 4. Wnt6 was not expressed in γδ T cells (Fig. 7b). Both Wnt8a and Wnt8b, however, were upregulated in γδ T cells, but not in DCs, from 3 and 4 day-immunized mice. When we analyzed Wnt8a and Wnt8b expression in TCRγδ$^+$CXCR5$^+$ vs. TCRγδ$^+$CXCR5$^-$ cells 4 days after immunization, we found that Wnt8b was highly upregulated in TCRγδ$^+$CXCR5$^+$ cells as compared to TCRγδ$^+$CXCR5$^-$ cells. Wnt8a was minimally expressed in both γδ T cell subtypes as compared to Wnt8b (Fig. 7c).

To investigate whether γδ T cells directly induce Ascl2 and thus Cxcr5 mRNA expression in CD4$^+$ T cells in vitro, we sorted TCRγδ$^+$CXCR5$^+$, TCRγδ$^+$CXCR5$^-$ or CD11c$^+$ DCs from WT mice 4 days after immunization, pulsed them with OVA$_{323-339}$ peptide and co-cultured with naïve CD4$^+$ T cells from OT-II-Foxp3-GFP mice. We found that TCRγδ$^+$CXCR5$^+$ cells and to a lesser extent CD11c$^+$ DCs, but not TCRγδ$^+$CXCR5$^-$ cells, induced both Ascl2 (36 h in co-culture) and Cxcr5 (72 h in co-culture) mRNA expression in CD4$^+$ T cells (Fig. 7d). However, CXCR5 protein expression was only observed on CD4$^+$ T cells cultured for 4 days in the presence of TCRγδ$^+$CXCR5$^+$ cells (Fig. 7e). Importantly, this effect was dependent on cell–cell contact because TCRγδ$^+$CXCR5$^+$ cells not loaded with OVA peptide or deficient for MHC-II did not induce CXCR5 expression on CD4$^+$ T cells (Fig. 7e; Supplementary Fig. 9b).

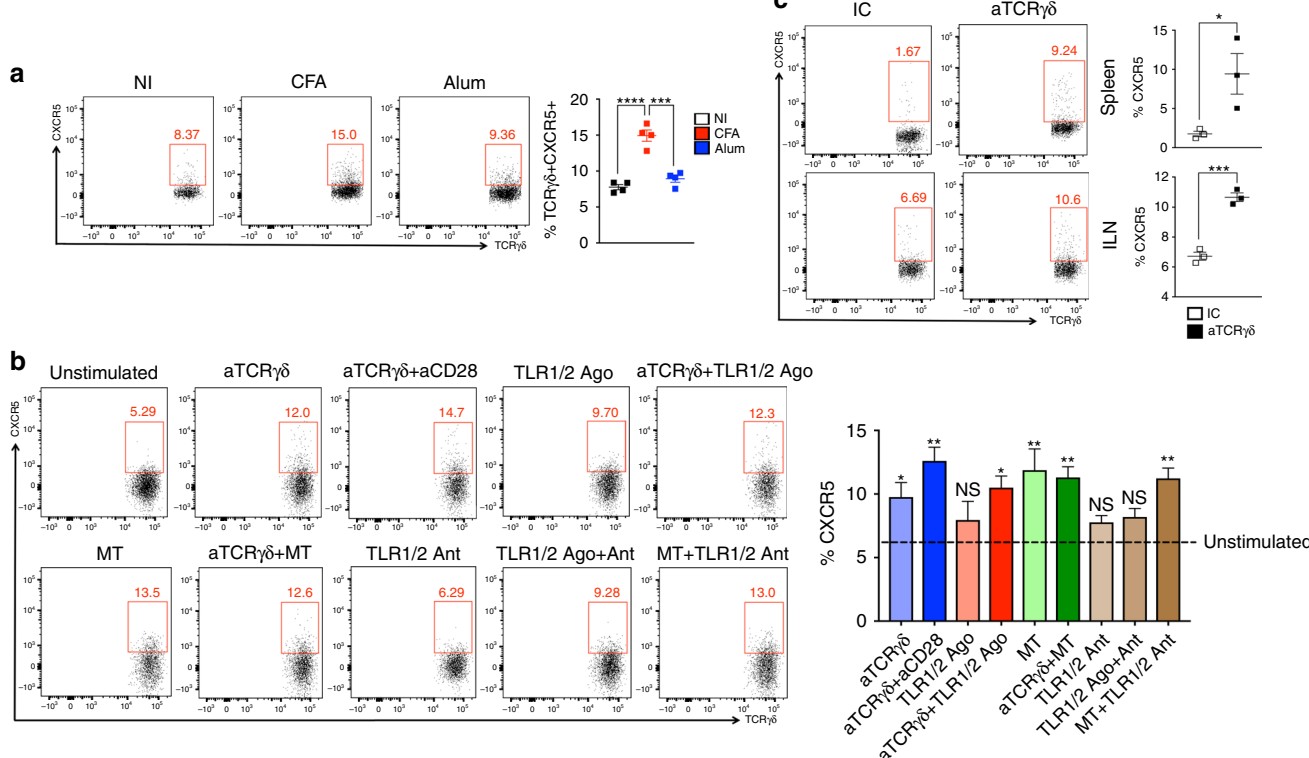

**Fig. 5** Mechanism of CXCR5 induction on γδ T cells. **a** Frequency of γδ T cells expressing CXCR5 in the dLN of non-immunized (NI) WT mice 7 days post either CFA or Alum immunization ($n = 4$ mice/group). **b** Frequency of γδ T cells expressing CXCR5 after 48 h of in vitro culture of $5 \times 10^4$ sorted CXCR5- γδ T cells from a pool of spleen and lymph nodes of 15 naïve WT mice in the presence of medium only (unstimulated); anti-TCRγδ monoclonal antibody (aTCRγδ; 2 μg ml$^{-1}$); anti-TCRγδ (2 μg ml$^{-1}$) + anti-CD28 (aCD28, 5 μg ml$^{-1}$) monoclonal antibodies; TLR1/2 agonist (TLR1/2 ago, 500 ng ml$^{-1}$); anti-TCRγδ (2 μg ml$^{-1}$) + TLR1/2 agonist (500 ng ml$^{-1}$); *Mycobacterium tuberculosis* (MT, 10 μg ml$^{-1}$); anti-TCRγδ (2 μg ml$^{-1}$) + MT (10 μg ml$^{-1}$); TLR1/2 antagonist (TLR1/2 ant, 0.5 μM); TLR1/2 agonist (500 ng ml$^{-1}$) + TLR1/2 antagonist (TLR1/2 ant, 0.5 μM) or MT (10 μg ml$^{-1}$) + TLR1/2 antagonist (0.5 μM) ($n = 3$ technical replicates/condition). **c** Frequency of γδ T cells expressing CXCR5 in the spleen and inguinal lymph node (ILN) from WT mice 24 h after treatment with 500 μg of anti-TCRγδ monoclonal antibody ($n = 3$ mice/group). These data are representative of two independent experiments. Data are shown as mean + SEM. One-way ANOVA (**a**, **b**) or Student's *t* test (**c**) were used. Conditions were compared with unstimulated cells. NS non-significant, $*p < 0.05$, $**p < 0.01$, $***p < 0.001$, $****p < 0.0001$

To determine whether Wnt ligands were involved in this process, we used the porcupine (PORCN) inhibitor Wnt-C59. PORCN is a membrane-bound O-acyltransferase responsible for endogenous Wnt ligand palmitoylation and its subsequent secretion from the cell[24]. Sorted TCRγδ$^+$CXCR5$^+$, TCRγδ$^+$CXCR5$^-$ or CD11c$^+$ DCs received Wnt-C59 at the same time they were pulsed with OVA$_{323–339}$ peptide. In this case, Wnt ligands were not released by the cells and consequently did not bind to their receptors on naïve CD4$^+$ T cells, which were added to the culture system on the next day. We found that CD4$^+$ T cells cultured in the presence of Wnt-C59-treated TCRγδ$^+$CXCR5$^+$ cells did not upregulate *Ascl2* or *Cxcr5* mRNA (Fig. 6d), and showed a 2-fold reduction in CXCR5 protein expression (Fig. 6e). No changes were observed in CD4$^+$ T cells co-cultured with either TCRγδ$^+$CXCR5$^-$ or CD11c$^+$ DCs (Fig. 6d, e). To investigate the role of Wnt8b in CXCR5 expression on CD4 T cells induced by TCRγδ$^+$CXCR5$^+$ cells, anti-Wnt8b monoclonal antibody was added together with naïve CD4 T cells. We found that anti-Wnt8b prevented CXCR5 induction on CD4 T in the same extent as did Wnt-C59 (Fig. 7e), confirming the crucial role of Wnt8b in the Tfh cell differentiation induced by TCRγδ$^+$CXCR5$^+$ cells. Of note, TCF-1, a key transcription factor downstream of Wnt and β-catenin pathway, was not required for Tfh cell differentiation induced by TCRγδ$^+$CXCR5$^+$ cells, because transgenic mice overexpressing TCF-1 and depleted of γδ T cells still required γδ T cells for CXCR5 induction on CD4 T cells. In fact, TCF-1 had

an opposite effect on Tfh cell differentiation, because its overexpression reduced the frequency of Tfh cells in immunized TCF-1 transgenic mice (Supplementary Fig. 9c).

To further confirm the ability of TCRγδ$^+$CXCR5$^+$ cells to induce CXCR5 expression on CD4$^+$ T cells in vitro, we stimulated TCRγδ$^+$CXCR5$^-$ and TCRγδ$^+$CXCR5$^+$ cells with plate-bound anti-CD3/anti-CD28 in the presence or not of Wnt-C59. Three days later, we collected the supernatant and added it to the plate containing naïve CD4$^+$ T cells under plate-bound anti-CD3/anti-CD28 stimulation. We found that supernatant from TCRγδ$^+$CXCR5$^+$ cells, but not from TCRγδ$^+$CXCR5$^-$ cells induced CXCR5 on CD4$^+$ T cells. This effect was prevented by adding supernatant from TCRγδ$^+$CXCR5$^+$ cells treated with Wnt-C59 (Fig. 8a). Taken together, these data indicate that Wnt/β-catenin pathway activation underlies the mechanism by which TCRγδ$^+$CXCR5$^+$ cells induce Tfh cells.

We next investigated whether γδ T cell-dependent CXCR5 expression on CD4$^+$ T cells occurred in vivo by pharmacologically inhibiting PORCN activity. It has been shown that the plasma concentration of Wnt-C59 after a single in vivo administration remained greater than 10-fold above the in vitro IC$_{50}$ for at least 16 h[25]. Thus, we treated naïve WT mice with either 10 mg/kg of Wnt-C59 or vehicle i.p. and 24 h later, total γδ T cells were isolated from spleen and lymph nodes and adoptively transferred into TCRδ$^{-/-}$ mice ($5 \times 10^5$ cells/mouse). Of note, we could not sort enough TCRγδ$^+$CXCR5$^+$ and TCRγδ$^+$CXCR5$^-$

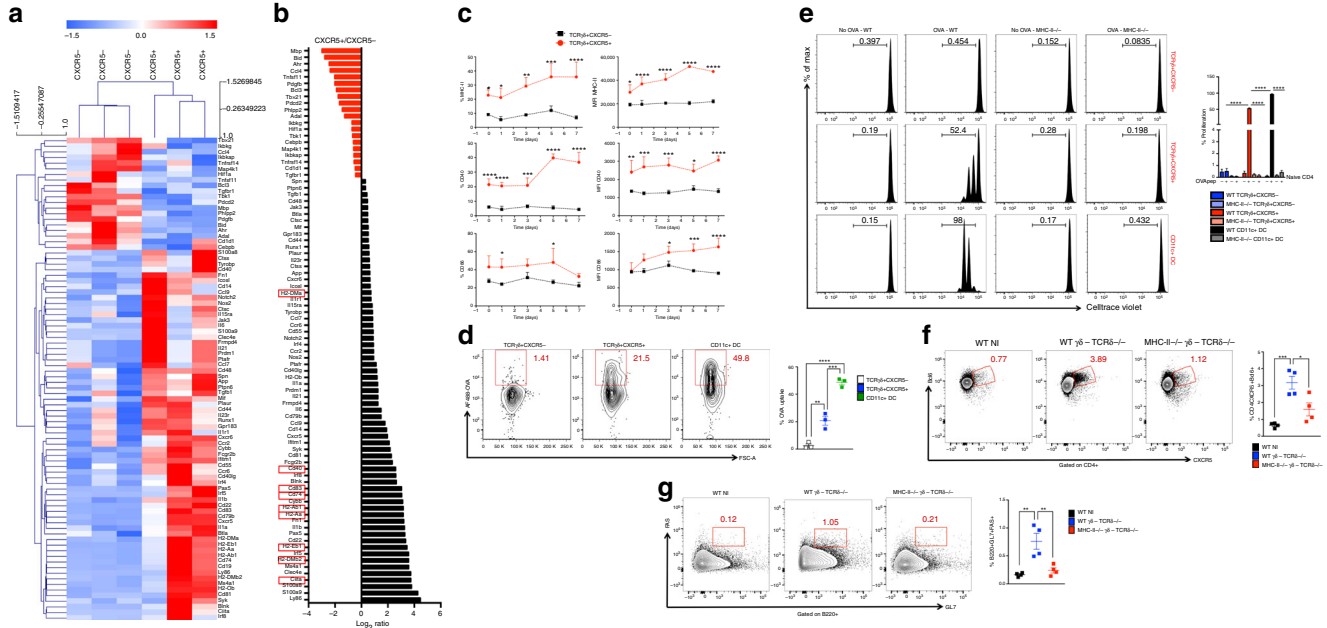

**Fig. 6** TCRγδ+CXCR5+ cells function as antigen-presenting cells. **a** Transcriptional signatures of TCRγδ+CXCR5− and TCRγδ+CXCR5+ cells sorted from dLN of WT mice 7 days after CFA immunization. Only statistically different ($p < 0.05$) mRNA levels are shown ($n$ = pooled cells from 5 mice/column). **b** Log2 ratio ($p < 0.05$) of mRNA levels from TCRγδ+CXCR5− and TCRγδ+CXCR5+ cells. Red squares indicate mRNAs related to antigen presentation function. **c** Frequency and median fluorescence intensity (MFI) of MHC-II, CD40, and CD86 expression on TCRγδ+CXCR5− and TCRγδ+CXCR5+ cells from dLN of WT mice before immunization (day 0) and 1, 3, 5, and 7 days after CFA immunization ($n$ = 5 mice/group). **d** Soluble ovalbumin (OVA) coupled to Alexa Fluor 488 (OVA-Alexa Fluor 488) endocytosed by TCRγδ+CXCR5−, TCRγδ+CXCR5+ and CD11c+ dendritic cells after 3 h of culture at 37 °C ($n$ = pooled cells from 12 mice). **e** Proliferation of CellTrace Violet-stained naïve CD4 T cells from OT-II-Foxp3-GFP mice co-cultured (3 days at 37 °C) with or without OVA323–339-loaded TCRγδ+CXCR5−, TCRγδ+CXCR5+ or CD11c+ dendritic cells (DC) from dLN of WT or MHC-II−/− mice 4 days after CFA immunization ($n$ = pooled cells from 10 mice/experiment). **f, g** In vivo Tfh cell induction (**f**) and germinal center formation (**g**) in dLNs of TCRδ−/− mice transferred intravenously with $5 \times 10^5$ γδ T cells from either WT or MHC-II−/− mice. Three days after transfer, recipient mice were immunized or not (WT NI) with CFA and 7 days thereafter sacrificed for flow cytometric analysis ($n$ = 4 mice/group, where γδ T cells are sorted from a pool of 10 mice/group). These data are representative of two independent experiments (**a–e**). Data are shown as mean + SEM. Pearson's correlation was used for hierarchical clustering in **a**. Student's $t$-test (**a–c**) or one-way ANOVA (**d–g**) were used. *$p < 0.05$, **$p < 0.01$, ***$p < 0.001$, ****$p < 0.0001$

cells to be separately transferred. We then immunized the mice with CFA and measured Tfh cells in the dLNs and OVA-specific antibodies in the sera 7 and 21 days later, respectively. We found that TCRδ−/− mice transferred with γδ T cells from Wnt-C59-treated mice had reduced frequencies of Tfh cells (Fig. 8b) and decreased OVA-specific IgM and IgG (Supplementary Fig. 9d) as compared to TCRδ−/− mice transferred with γδ T cells from vehicle-treated mice. Of note, CXCR5 expression on γδ T cells was similar between groups (Fig. 8c). Thus, these data indicate that Wnt ligands secreted by γδ T cells are important for the Tfh cell induction after immunization.

## Discussion

γδ T cells have a wide range of biologic function, from fighting pathogens and tumors[26] to regulating mucosal immunity including intestinal inflammation[27] and oral tolerance[28–30]. Importantly, γδ T cells have also been shown to participate in the humoral immune response by helping B cells form GC and differentiate into plasma cells[10,31]. However, it is not known whether γδ "Tfh-like" cells exist or whether γδ T cells communicate directly with B cells or affect Tfh cell development.

It has been shown that even in the absence of αβ T cells, B cells can expand and secrete T cell-dependent antibodies that react to self-antigens[8]. Moreover, it is now known that in non-immunized mice or in mouse models of allergic diseases, Vγ1+ γδ T cells favor antibody production whereas Vγ4 γδ T cells suppress it[10,15,16]. Accordingly, in the present study, we found that the

participation of γδ T cells in antibody production following CFA immunization relies on the crosstalk between Vγ1 and Vγ4 γδ T cell subsets. Vγ1+ γδ T cells promoted antibody production, whereas Vγ4+ γδ T cells suppressed it. Consistent with the role of Vγ1+ γδ T cells in promoting humoral immune response, we found that Tfh cell frequency was higher in Vγ4+ γδ T cell-depleted mice than in WT mice immunized with CFA/OVA, but not with Alum/OVA. Moreover, CFA, but not Alum, induced the expression of CXCR5 and MHC-II exclusively on Vγ1+ γδ T cells, which we have shown to play a crucial role in Tfh cell differentiation induced by γδ T cells. This lack of γδ T cell activation following Alum immunization may explain the differential involvement of γδ T cells on CFA- vs. Alum-induced antibody production. CFA-activated γδ T cells acquire antigen presentation functions and secrete Wnt ligands in order to initiate the Tfh cell differentiation program.

We found that γδ T cells did not rescue the hypoglobulinemia observed in TCRβ−/− mice, indicating that γδ T cells require αβ T cells to help B cells produce OVA-specific antibodies. However, because we did not investigate the involvement of Vγ1 and Vγ4 γδ T cell subsets in a TCRβ−/− background, we cannot exclude the possibility that immunoglobulin subclasses such as IgM, IgG2b, and IgG3, may be controlled by γδ T cells in an αβ T cell-independent fashion, as previously described[10]. In this context, the low levels of antibodies observed in TCRβ−/− mice could be a consequence of the inhibitory effect of Vγ4 γδ T cells[10].

γδ T cells have been shown to participate in the pathogenesis of SLE through their antigen-presenting function, production of

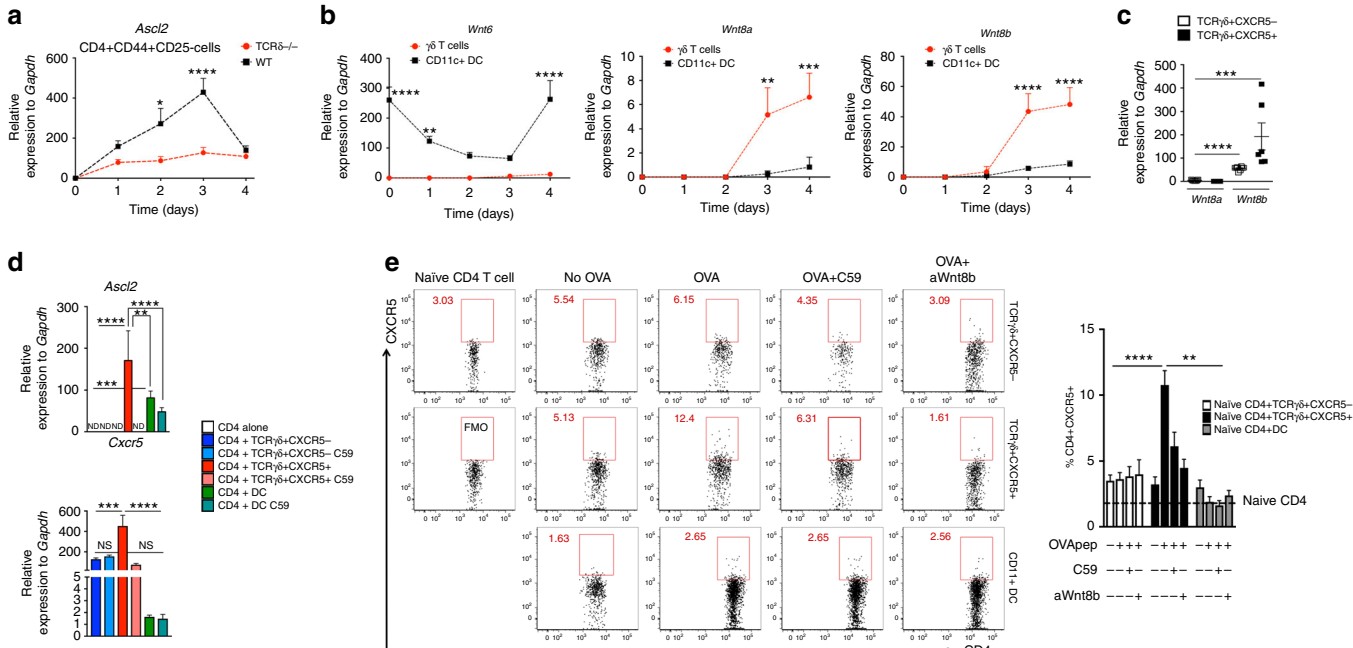

**Fig. 7** TCRγδ+CXCR5+ cells release Wnt ligands to induce CD4+CXCR5+ cells. **a** Quantitative RT-PCR analysis of *Ascl2* mRNA from dLN-sorted CD4+CD44+CD25− T cells of WT and TCRδ−/− mice before immunization (day 0) and 1, 2, 3, and 4 days after CFA immunization (*n* = pooled cells from 5 mice/group/time point). Data are from three experiments. **b** *Wnt6*, *Wnt8a*, and *Wnt8b* from dLN-sorted γδ T cells and CD11c+ dendritic cells (DC) of WT mice before immunization (day 0) and 1, 2, 3, and 4 days after CFA immunization (*n* = pooled cells from 5 mice/group/time point). Data are from three experiments. **c** *Wnt8a* and *Wnt8b* from dLN-sorted TCRγδ+CXCR5− and TCRγδ+CXCR5+ cells of WT mice 4 days after CFA immunization (*n* = pooled cells from 5 mice/group). Data were combined from **b**. **d** Quantitative RT-PCR analysis of *Ascl2* (36 h in co-culture) and *Cxcr5* (72 h in co-culture) mRNAs from OT-II-Foxp3-GFP mouse-sorted naïve CD4+ T cells cultured in the presence of OVA323–339-loaded TCRγδ+CXCR5−, TCRγδ+CXCR5+cells or CD11c+ DCs treated or not with the porcupine inhibitor Wnt-C59 (C59; 1 μM) for 3 days at 37 °C. γδ T cells and DCs were sorted from WT mice (pooled from 15 mice/experiment) 4 days after CFA immunization. Data are from three experiments. **e** CXCR5 induction on naïve CD4 T cells from OT-II-Foxp3-GFP mice co-cultured (3 days at 37 °C) with or without OVA323–339-loaded TCRγδ+CXCR5−, TCRγδ+CXCR5+ or CD11c+ DCs in the presence or not of either 1 μM of Wnt-C59 (C59) or 20 μg ml−1 of anti-Wnt8b monoclonal antibody (aWnt8b) from WT mice 4 days after CFA immunization (*n* = pooled cells from 15 mice/experiment). Data are from at least two experiments. Data are shown as mean + SEM. Two-way ANOVA (**a**–**b**) or one-way ANOVA (**c**–**e**) were used. ND not detected, *$p < 0.05$, **$p < 0.01$, ***$p < 0.001$, ****$p < 0.0001$

proinflammatory cytokines, interaction with regulatory T cells, and promoting antibody production by providing B cell help[32]. Consistent with this, we found a reduction in anti-nuclear and anti-complement antibodies, which are known to participate in the pathogenesis of SLE[17], both in naïve and pristane-induced lupus TCRδ−/− mice. This correlated with milder glomerulonephritis in lupus-induced TCRδ−/− mice. Importantly, these effects were accompanied by a reduced expression of Tfh cells in TCRδ−/− mice, suggesting that diminished autoantibody production in these mice relies on their inability to initiate a proper Tfh cell differentiation program.

We hypothesized that γδ T cells play either a direct role in B cell activation, behaving as γδ Tfh-like cells or an indirect role by helping Tfh cell differentiation and function. Our findings demonstrate that although both OVA-specific antibody and autoantibody serum levels were reduced in TCRδ−/− mice as compared to WT mice upon CFA immunization, the B cell compartment was intact, suggesting that a direct contact between γδ T and B cells is unlikely. Consistent with this, even though a subtype of γδ T cell expressed CXCR5, PD-1, and ICOS, it failed to express Bcl6, produce IL-21 and had only marginal expression of CD40L, which are crucial factors for direct B cell help[33]. TCRδ−/− mice showed reduced Tfh cell frequencies upon CFA immunization as compared to WT mice. This led to impaired GC formation. Moreover, Tfh cells from TCRδ−/− mice had low expression of CD40L, indicating that the Tfh compartment in immunized TCRδ−/− mice is not only reduced, but also

dysfunctional in providing help to B cells. Notably, 5 months after pristane injection, TCRδ−/− mice showed decreased Tfh cell frequencies as compared to WT mice, which is consistent with the reduced levels of autoantibodies in TCRδ−/− mice. Interestingly, it has been recently shown that γδ T cells from zebrafish, by functioning as APCs, similar to what we observed for murine γδ T cells, activated CD4+ T cells that began to express CD40L and help B cells produce antibodies[34]. Thus, the crosstalk between γδ T cells and CD4+ T cells seems to be a conserved evolutionary mechanism in the context of humoral immunity.

Tfh cell differentiation is a multistage and multifactorial process with several signals provided by APCs. A crucial step in the Tfh cell differentiation is the induction of Bcl6, which maintains the Tfh cell program by avoiding further Tfh cell differentiation into Th1, Th2, or Th17 cells. However, it is not clear how the Tfh cell program begins. In this regard, Liu et al. demonstrated that CXCR5 expression on CD4+ T cells occurs previously to Bcl6 induction and is mediated by the transcription factor Ascl2[7]. Thus, although Bcl6 has been shown to be essential for Tfh cell development, it does not regulate the initial steps of Tfh cell program.

Ascl2 can be induced by Wnt agonists[7], indicating the involvement of the β-catenin pathway in Tfh cell generation. However, we found that Ascl2 acts independently of TCF-1, a key transcription factor downstream of Wnt and β-catenin pathway, in initiating the Tfh cell program because transgenic mice overexpressing TCF-1 and depleted of γδ T cells still required γδ

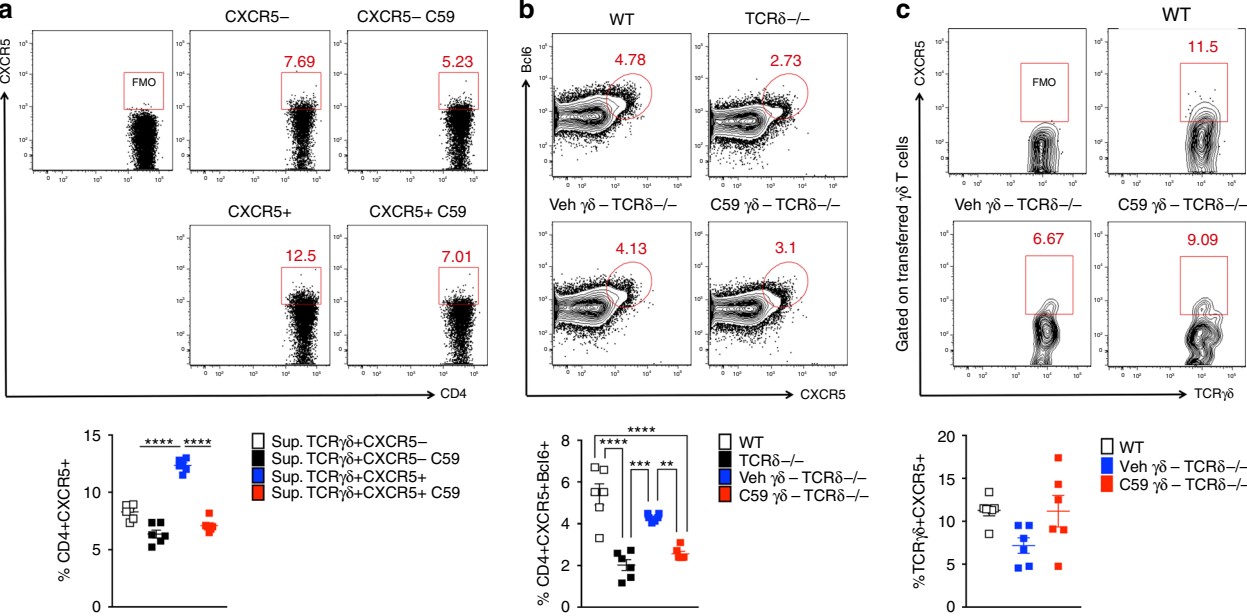

**Fig. 8** TCRγδ+CXCR5+ cells induce Tfh cells by releasing Wnt ligands. **a** In vitro CXCR5 induction on naïve CD4+ T cells cultured in the presence of supernatants from TCRγδ+CXCR5− or TCRγδ+CXCR5+ cells, treated or not with the porcupine inhibitor Wnt-C59 (C59; 1 μM), sorted from WT mice 4 days after CFA immunization and activated with anti-CD3ε and anti-CD28 (both at 2 μg ml⁻¹) for 3 days at 37 °C. Naïve CD4+ T cells, sorted from naïve WT mice, cultured with the described supernatants were also stimulated with anti-CD3ε and anti-CD28 (both at 2 μg ml⁻¹) for 3 days at 37 °C ($n$ = pooled cells from 15 mice). **b**, **c** In vivo CXCR5 and Bcl6 induction in CD4+ T cells (**b**) and CXCR5 expression on γδ T cells (**c**) from dLN of TCRδ⁻/⁻ mice transferred with γδ T cells from WT mice treated with either 10 mg kg⁻¹ of Wnt-C59 (C59) or vehicle (Veh). WT mice were sacrificed 24 h after Wnt-C59 or vehicle treatment and a total of $5 \times 10^5$ γδ T cells were intravenously injected into TCRδ⁻/⁻ mice, which were immunized with CFA and sacrificed 7 days later ($n$ = 6 mice/group, combined from three independent experiments where γδ T cells are sorted from a pool of 10 mice/group/experiment). These data are representative of two to three independent experiments. Data are shown as mean + SEM. One-way ANOVA was used. *$p < 0.05$, ***$p < 0.001$

T cells for CXCR5 induction on CD4 T cells. In fact, TCF-1 had an opposite effect on Tfh cell differentiation, because its overexpression reduced the frequency of Tfh cells in immunized TCF-1 transgenic mice. This is consistent with an independent mechanism between TCF-1 and Ascl2 in β-catenin responsive gene activation[35]. Importantly, we found that CD4+CD44+ CD25− T cells from immunized TCRδ⁻/⁻ mice had substantially less *Ascl2* mRNA as compared to WT mice, suggesting that γδ T cells are involved in Ascl2 induction in CD4+ T cells. Indeed, γδ T cells expressed Wnt8a and Wnt8b, which directly correlated with *Ascl2* mRNA induction kinetics in CD4+ T cells. We found that these Wnt ligands were increased in a subset of γδ T cells that expressed CXCR5. TCRγδ+CXCR5+ cells functioned as APCs and directly induced Ascl2 and CXCR5 on naïve CD4+ T cell in vitro and Tfh cells (CD4+CXCR5+Bcl6+) in vivo. This effect was mediated by Wnt ligand secretion because inhibiting the enzyme responsible for Wnt release, PORCN, abrogated both Ascl2 and CXCR5 expression in CD4+ T cells. Notably, Wnt8b plays a major role in CXCR5 expression on CD4 T cells since blocking Wnt8b with a monoclonal antibody prevented this effect. Thus, TCRγδ+CXCR5+ cells, by functioning as APCs and releasing Wnt ligands, induce Ascl2 in CD4+ T cells that in turn initiates the Tfh cell program. The fact that γδ T cells control antibody production by presenting antigen and inducing Tfh cell differentiation does not exclude the crucial role of DCs in Tfh cell development. Because TCRγδ+CXCR5+ cells do not express ICOSL, they cannot induce Bcl6 in CD4+CXCR5+ T cells that, in the absence of DCs, would abort Tfh cell differentiation[5].

CXCR5-expressing γδ T cells could be observed even in non-immunized mice, and immunization increased the number of γδ T cells in the T cell zone, but also in the follicles of dLNs, which is consistent with the upregulation of CXCR5 on γδ T cells. The role

of γδ T cells in the follicle is not known, particularly taking into consideration our findings that γδ T cells do not express CD40L nor produce IL-21, critical factors for B cell activation and maturation[36,37]. However, the presence of γδ T cells within the follicles may be related to their ability to secrete Wnt ligands, which have been shown to protect germinal center B cells from rapid apoptosis[38]. The frequency of TCRγδ+CXCR5+ cells increased 1 day after immunization, which is in contrast with Tfh cells that gradually upregulated CXCR5. This suggests that γδ T cells have a rapid mechanism for CXCR5 induction. Consistent with this, we found that γδTCR activation, but not toll-like receptor (TLR) 1/2, played a crucial role in CXCR5 upregulation on γδ T cells. CFA, but not Alum, induced CXCR5-expressing γδ T cells in vivo, particularly on the Vγ1+ γδ T cell subset, and this effect was likely mediated by the *M. tuberculosis* present in the CFA. Accordingly, MT contains heat shock protein 65 (Hsp65), that can directly activate the γδTCR from Vγ1+ γδ T cells[39]. Notably, pristane injection also induced CXCR5 upregulation on γδ T cells, again demonstrating that pathogen recognition receptors (PRRs) other than TLR1/2 may also induce CXCR5 expression on γδ T cells.

In summary, our findings help explain the poorly understood mechanism by which γδ T cells participate in the humoral immunity. We propose the following: upon arrival in secondary lymphoid organs, antigens directly activate γδ T cells by binding to TCR and/or PRRs, which induce CXCR5 expression on a subset of γδ T cells. TCRγδ+CXCR5+ cells migrate toward the follicle and present antigens to naïve CD4+ T cells via MHC-II. Wnt ligands are released during this process to induce Ascl2 and initiate the Tfh cell program. Ascl2 leads to CXCR5 upregulation on CD4+ T cells[7]. DCs, which also present antigens to CD4+ T cells, provide co-stimulatory signals that lead to Bcl6

expression, which terminates Tfh cell differentiation. These new generated Tfh cells activate cognate antigen-expressing B cells at the B:T cell border by providing CD40L and IL-21 signals, resulting in the generation of high-affinity antibody-secreting plasma cells. Our results raise the possibility that patients affected by antibody-mediated autoimmune conditions may benefit from γδ T cell modulation as a component for treatment of their disease.

## Methods

**Mice.** Male and female, 6–10 week-old on B6 genetic background mice were used in this study. C57BL/6J wild type, MHC-II$^{-/-}$, CXCR5$^{-/-}$, TCRβ$^{-/-}$, TCRγδ-GFP, and TCRδ$^{-/-}$ mice were purchased from the Jackson Laboratory. Tcf7L (TCF-1) Tg mice were kindly provided by Dr. Ana Anderson from Brigham and Women's Hospital, Harvard Medical School, Boston, MA, USA. OT-II-Foxp3-GFP mice were housed in a conventional specific pathogen-free facility at the Harvard Institutes of Medicine and/or at the Building for Transformative Medicine according to the animal protocol with the full knowledge and permission of the Standing Committee on Animals at Harvard Medical School and Brigham and Women's Hospital. Three to ten mice were used per group and every experiment was reproduced at least twice, which is commonly used for in vivo studies to alleviate unnecessary animal suffering. No animal was excluded from analysis and no randomization was performed.

**FACS and intracellular cytokine staining.** In the experiments where sorted cells were required, a pool of cells from inguinal lymph nodes (ILN) of C57BL/6, TCRδ$^{-/-}$ or OT-II-Foxp3-GFP mice were first enriched using CD4 microbeads or TCRγδ isolation kit (all from Miltenyi Biotec). Then, naïve (CD4$^+$CD62L$^+$CD44$^-$Foxp3$^-$) or memory (CD4$^+$CD62L$^-$CD44$^+$CD25$^-$) cells were sorted (FACS Aria II, BD Bioscience) with peridinin chlorophyll protein (PerCP)–conjugated anti-CD4 (RM4–5; 1:250), allophycocyanin (APC)-conjugated anti-CD62L (MEL-14; 1:250) and phycoerythrin (PE)-conjugated anti-CD44 (IM7; 1:500; all from BioLegend). TCRγδ$^+$CXCR5$^+$ and TCRγδ$^+$CXCR5$^-$ cells were sorted with Alexa Fluor 700 (AF700)-conjugated anti-CD3ε (eBio500A2; 1:100; eBioscience), APC-conjugated anti-TCRγδ (eBioGL3; 1:100; eBioscience) and PE-Cy7-conjugated anti-CXCR5 (L138D7; 1:100; Biolegend). For TCRγδ cell sorting, AF488-conjugated anti-CD11c (N418; 1:100; Biolegend) was used to exclude any remaining CD11c$^+$ cells to prevent dendritic cell contamination. Dead cells were also excluded based on 7-AAD (BD Bioscience) staining. In some experiments, antigen-presenting cells were first enriched using CD11c microbeads (Miltenyi Biotec) and sorted for CD11c$^+$ cells with exclusion of dead cells based on 7-AAD staining. Gating strategies for cell sorting is shown in Supplementary Fig. 10. For intracellular cytokine staining, cells were first stimulated for 4 h with PMA (phorbol 12-myristate 13-aceate; 50 ng ml$^{-1}$; Sigma-Aldrich) and ionomycin (1 μM; Sigma-Aldrich) and a protein-transport inhibitor containing monensin (1 μg ml$^{-1}$ GolgiStop; BD Biosciences) before detection by staining with antibodies. Surface markers were stained for 25 min at 4 °C in Mg$^{2+}$ and Ca$^{2+}$ free HBSS with 2% FCS, 0.4% EDTA (0.5 M) and 2.5% HEPES (1 M) then were fixed in Cytoperm/Cytofix (eBioscience), permeabilized with Perm/Wash Buffer (eBiosciences), Flow-cytometric acquisition was performed on a Fortessa (BD Biosciences) by using DIVA software (BD Biosciences) and data were analyzed with FlowJo software versions 9.9 or 10.1 (TreeStar Inc.). Intracellular staining antibodies used: BV-421-anti-IFN-γ (XMG1.2; 1:300; Biolegend), PE-Cy7-anti-IFN-γ (XMG1.2; 1:200; eBioscience), FITC-anti-IL-17A (eBio17B7; 1:100; eBioscience), PE-anti-IL-21 (FFA21; 1:100; eBioscience). Other antibodies included: BV605-anti-CD4 (RM4.5; 1:300; BD Bioscience), BV421-anti-CD4 (GK1.5; 1:300; Biolegend), BV421-anti-TCRγδ (GL3; 1:100; Biolegend), PE-anti-TCRγδ (GL3; 1:100; Biolegend); AF700-anti-CD19 (eBio1D3; 1:100; eBioscience), AF488-anti-B220 (RA3-6B2; 1:100; BD Biosciences), AF647-anti-B220 (RA3-6B2; 1:100; BD Biosciences), PE-Cy5-anti-B220 (RA3-6B2; 1:100; BD Biosciences), PE-Cy7-anti-IgM (RMM-1; 1:200; Biolegend), APC-anti-IgD (11-26c.2a; 1:200; Biolegend), PE-anti-IgD (11–26; 1:100; BD Biosciences), BV605-anti-CD138 (281-2; 1:100; BD Biosciences), BV421-anti-CD43 (S7; 1:100; BD Biosciences), PE-anti-BP-1 (BP-1; 1:100; BD Biosciences), AF700-anti-CD24 (M1/69; 1:100; BD Biosciences), PE-Cy5-anti-IL-21R (4A9; 1:100; Biolegend), APC-anti-ICOS (C398.4A; 1:100; eBioscience), PE-anti-BCL6 (IG19E/A8; Biolegend), PerCP-efluor710-anti-BCL6 (BCL-DWN; 1:200; eBioscience), APC-anti-CD40L (MR1; 1:100; eBioscience), APC-anti-GL7 (GL7; 1:100; Biolegend), AF488-anti-CD95 (15A7; 1:100; eBioscience), PE-Cy7-anti-I-A/I-E (M5/114.15.2; 1:200; Biolegend), AF647-anti-I-A/I-E (M5/114.15.2; 1:200), BV650-anti-I-A/I-E (M5/114.15.2; 1:200; Biolegend), APC-anti-CD11c (N418; 1:100; eBioscience), AF488-anti-CD11c (N418; 1:100; Biolegend), PE-Cy7-anti-CD11c (N418; 1:100; eBioscience), AF700-anti-CD86 (GL-1; 1:100; Biolegend), AF488-anti-CD86 (GL-1; 1:100; Biolegend), PE-anti-CD40 (3/23; 1:100), PE-anti-TCR Vβ5.1/5.2 (MR9-4; Biolegend), APC-anti-Vγ1.1/Cr4 (2.11; 1:200; Biolegend), PE-anti-Vγ2 ("anti-Vγ4"; UC3-10A6; 1:200; Biolegend).

**Immunization.** Mice were immunized subcutaneously (s.c.) in the flanks with 100 μl (per flank) of either Complete Freud Adjuvant (CFA) enriched with 4 mg ml$^{-1}$ of *M. tuberculosis* H37RA (Difco) plus PBS or CFA enriched with 4 mg ml$^{-1}$ of *M. tuberculosis* H37RA plus OVA (50 μg per mouse; Sigma). In some experiments mice were also immunized (s.c.) in the flanks (100 μl per flank) or i.p. (200 μl) with OVA (50 μg per mouse) adsorbed in 1 mg of Alum (Aluminum hydroxide hydrate, Sigma). To investigate the involvement of different subtypes of γδ T cells on their ability to control the humoral immune response, 200 μg of anti-Vγ2 mAb ("anti-Vγ4"; UC3-10A6; BioXCell) or isotype control (BioXCell) were injected i.p. three days before either CFA or Alum immunization and three days before OVA booster dose.

**Autoantibody induction by pristane.** Eight-week-old C57BL/6 and TCRδ$^{-/-}$ mice received a single i.p. injection of 0.5 ml of pristane (Sigma). Sera were collected before pristane injection and 3 months thereafter for autoantibody analysis. Five months after pristane injection, mice were sacrificed and kidneys removed for histopathology investigation as well as spleens taken for FACS analysis.

**Serum antibody levels.** Eight-week-old C57BL/6, TCRβ$^{-/-}$, and TCRδ$^{-/-}$ mice were first immunized with OVA emulsified in CFA containing 4 mg ml$^{-1}$ of *M. tuberculosis* H37RA (Difco), or adsorbed in Alum as described above. Fourteen days later, mice received an i.p. injection of OVA (50 μg) in PBS and were sacrificed 7 days thereafter for sera and intestinal lavage fluids collection. Ready-SET-Go! ELISAs (eBioscience) for detecting total IgG antibodies and subclasses were performed according to the manufacturer's instructions. Sera were added at starting dilution 1:10,000, followed by 2-fold serial dilutions. For secretory IgA (sIgA) detection, intestinal lavage fluids were collected by flushing the colon with cold PBS, spun down for 20 min, 2000 rpm at 4 °C, and supernatants used for sIgA detection with starting dilution 1:100, followed by 2-fold serial dilutions. For anti-OVA antibody titers, 96-well plates (Nunc, Roskild, Denmark) were coated overnight with 20 μg ml$^{-1}$ OVA solution in coating solution (eBioscience) at 4 °C. Ready-SET-Go! ELISAs (eBioscience) for detecting total IgG antibodies and subclasses were then performed according to the manufacturer's instructions. Sera or intestinal lavage fluids were added at starting dilution 1:100, followed by 2-fold serial dilutions. Samples were read using a Tecan microplate reader (Tecan, Infinite 200 PRO) and processed using i-Control microplate reader software (Tecan).

**Autoantigen microarray.** Sera collected from both naïve wild type and TCRδ$^{-/-}$ mice were sent to Genomics and Microarray Core Facility at UT Southwestern Medical Center (Dallas, TX) for analysis, using the autoantigen microarray super panel (128 autoantigens). Briefly, samples were treated with DNAse I, diluted 1:50, and incubated with autoantigen array. The autoantibodies binding to the antigens on the array were detected with Cy3 labeled anti-IgG and Cy5 labeled anti-IgM and the arrays were scanned with GenePix® 4400A Microarray Scanner. The images were analyzed using GenePix 7.0 software to generate GPR files. The averaged net fluorescent intensity (NFI) of each autoantigen was normalized to internal controls (IgG or IgM).

**γδ T cell depletion.** γδ T cells were depleted by injecting a single dose of 250 μg i.p. of anti-TCRγδ depleting antibody (LEAF purified anti-TCRγδ, UC7-13D5; Biolegend).

**CXCR5 induction on γδ T cells.** $5 \times 10^4$ sorted CXCR5$^-$ γδ T cells from a pool of spleen and lymph nodes of 15 naïve WT mice were cultured for 48 h at 37 °C in the presence of medium only (unstimulated); anti-TCRγδ monoclonal antibody (UC7-13D5; Biolegend; 2 μg ml$^{-1}$); anti-TCRγδ (2 μg ml$^{-1}$) + anti-CD28 (37.52; BD Biosciences; 5 μg ml$^{-1}$) monoclonal antibodies; TLR1/2 agonist (Pam3CSK4 (Invivogen), 500 ng ml$^{-1}$); anti-TCRγδ (2 μg ml$^{-1}$) + TLR1/2 agonist (500 ng ml$^{-1}$); *M. tuberculosis* (Difco, 10 μg ml$^{-1}$); anti-TCRγδ (2 μg ml$^{-1}$) + MT (10 μg ml$^{-1}$); TLR1/2 antagonist (CU-CPT22 (Millipore Sigma), 0.5 μM); TLR1/2 agonist (500 ng ml$^{-1}$) + TLR1/2 antagonist (TLR1/2 ant, 0.5 μM) or MT (10 μg ml$^{-1}$) + TLR1/2 antagonist (0.5 μM). CXCR5 expression on γδ T cells was analyzed by flow cytometry.

For in vivo study of CXCR5 expression on γδ T cells, 500 μg per mouse of anti-TCRγδ monoclonal antibody (UC7-13D5; Biolegend) was administered i.p. and 24 h later CXCR5 expression on γδ T cells was analyzed by flow cytometry.

**Uptake and presentation assays.** CD11c$^+$ dendritic cells and TCRγδ$^+$CXCR5$^+$, TCRγδ$^+$CXCR5$^-$ cells were first enriched using CD11c microbeads or TCRγδ isolation kit (all from Miltenyi Biotec) and sorted. For TCRγδ sorting, AF488-conjugated anti-CD11c (N418; 1:100; Biolegend) was used to exclude any remaining CD11c$^+$ cells in order to prevent dendritic cell contamination. For uptake assay, TCRγδ$^+$CXCR5$^-$, TCRγδ$^+$CXCR5$^+$ and CD11c$^+$ dendritic cells were incubated for 3 h at 37 °C with 50 μg ml$^{-1}$ of ovalbumin (OVA) coupled to Alexa Fluor 488 (Invitrogen) in a 96-well round-bottom plate. After incubation, cells were collected, thoroughly washed and analyzed by flow cytometry. For in vitro presentation assay, cells were first incubated overnight at 37 °C with 50 μg ml$^{-1}$ of either OVA$_{323–339}$ peptide (Invivogen), OVA protein (Sigma) or medium only (unloaded cells as control) in a 96-well round-bottom plate. On the next day, cells were thoroughly washed and incubated at 1:2 ratio (antigen-presenting cells:

responder cells) with sorted naïve (CD4$^+$CD62L$^+$CD44$^-$Foxp3$^-$) cells from OT-II-Foxp3-GFP mice previously stained with CellTrace Violet dye (Invitrogen) for 4 days. Proliferation was then analyzed by flow cytometry.

**In vitro CXCR5 induction on CD4$^+$ T cells by γδ T cells**. CD11c$^+$ dendritic cells, TCRγδ$^+$CXCR5$^+$ and TCRγδ$^+$CXCR5$^-$ cells from WT mice were first enriched using CD11c microbeads or TCRγδ isolation kit (all from Miltenyi Biotec) and sorted. For TCRγδ sorting, AF488-conjugated anti-CD11c (N418; 1:100; Biolegend) was used to exclude any remaining CD11c$^+$ cells in order to prevent dendritic cell contamination. Cells were then incubated overnight at 37 °C with 50 µg ml$^{-1}$ of either OVA$_{323-339}$ peptide, OVA protein or medium only (unloaded cells as control) or 50 µg ml$^{-1}$ of OVA$_{323-339}$ peptide plus either 1 µM of the porcupine (PORCN) inhibitor (Wnt-C59, Tocris) or 20 µg ml$^{-1}$ of anti-Wnt8b monoclonal antibody (LSBio) in a 96-well round-bottom plate. On the next day, cells were thoroughly washed and incubated at 1:2 ratio (antigen-presenting cells: responder cells) with sorted naïve (CD4$^+$CD62L$^+$CD44$^-$Foxp3$^-$) cells from OT-II-Foxp3-GFP mice previously stained with CellTrace Violet dye (Invitrogen) for 4 days. CXCR5 expression on CD4 T cells was then analyzed by flow cytometry.

**CXCR5 induction on CD4$^+$ T cells by γδ T cell supernatant**. TCRγδ$^+$CXCR5$^+$ and TCRγδ$^+$CXCR5$^-$ cells from WT mice were first enriched using TCRγδ isolation kit (all from Miltenyi Biotec) and sorted. AF488-conjugated anti-CD11c (N418; 1:100; Biolegend) was used to exclude any remaining CD11c$^+$ cells in order to prevent dendritic cell contamination. Cells were then stimulated with plate-bound anti-CD3 (145-2C11; Biolegend) and anti-CD28 (PV-1; Bioxcell), both at 2 µg ml$^{-1}$, for 3 days at 37 °C in a 96-well round-bottom plate. Wnt-C59 at 1 µM was added to some wells to prevent Wnt ligand release. At the 3rd day of culture, supernatants were collected and 100 µl added to naïve (CD4$^+$CD62L$^+$CD44$^-$Foxp3$^-$) cells in a 96-well round-bottom plate and stimulated with plate-bound anti-CD3 and anti-CD28 (both at 2 µg ml$^{-1}$) at 37 °C. Three days later, CXCR5 expression on CD4 T cells was analyzed by flow cytometry.

**In vivo γδ T cell transfer**. To investigate whether CXCR5 expression and antigen presentation function of γδ T cells were involved in the ability of γδ T cells to induce Tfh cell differentiation in vivo, γδ T cells from spleen and lymph nodes of WT C57BL/6J, CXCR5$^{-/-}$ and MHC-II$^{-/-}$ mice were first enriched using TCRγδ isolation kit (all from Miltenyi Biotec) and 5 × 10$^5$ sorted γδ T cells were intravenously transferred into TCRδ$^{-/-}$ mice. Three days after transfer, mice were immunized with CFA and 7 days later draining (inguinal) lymph nodes were collected for FACS analysis. To investigate the involvement of Wnt ligands secretion by γδ T cells on their ability to induce Tfh cell differentiation, WT C57BL/6J mice were treated with either 10 mg/kg of Wnt-C59 (Tocris) or vehicle (3.5% DMSO in PBS) intraperitoneally and 1 day later, animals were sacrificed and spleens and lymph nodes collected. γδ T cells were first enriched using TCRγδ isolation kit (all from Miltenyi Biotec) and 5 × 10$^5$ sorted γδ T cells were intravenously transferred into TCRδ$^{-/-}$ mice. One day after transfer, mice were immunized with CFA and 7 days later draining (inguinal) lymph nodes were collected for FACS analysis.

**Whole lymph node confocal microscopy**. TCRγδ-GFP and TCRγδ$^{-/-}$ mice were injected s.c. in the inner thigh with a mix of 20 µl of BV421-anti-CD4 (GK1.5; 0.2 mg ml$^{-1}$; Biolegend), 20 µl of AF647-anti-B220 (RA3-6B2; 0.2 mg/ml; BD Bioscience) and PBS in a total volume of 100 µl/thigh. In some experiments, either 20 µl of PE-anti-CXCR5 (L138D7; 0.2 mg ml$^{-1}$; Biolegend) or 20 µl of PE-anti-GL7 (GL7; Biolegend) were also injected in combination with BV421-anti-CD4 and AF647-anti-B220. Three hours later, mice were sacrificed and inguinal lymph nodes removed for confocal microscopy. Lymph nodes were placed on a glass microscopy slide with PBS and imaged at 10× or 20× with Zeiss 710 confocal microscope.

**Histology**. Kidneys were excised from animals 5 months after pristane treatment and immediately fixed in 10% neutral buffered formalin. Samples were then embedded in paraffin and 5 µm were cut and stained with hematoxylin and eosin. Slides were loaded into an Aperio ScanScope XT (Aperio), scanned via the semi-automated method and checked for image quality using visual inspection. By using the eSlide Manager (Aperio), sections were evaluated for histopathological changes, such as glomerulus enlargement (diameter of 50 glomeruli per mouse were measured) and cellularity (number of nuclei per glomerulus in a total of 50 glomeruli per mouse).

**Real-time PCR**. Memory CD4$^+$CD44$^+$CD25$^-$ cells, CD11c$^+$ dendritic cells, total γδ T cells, TCRγδ$^+$CXCR5$^+$ and TCRγδ$^+$CXCR5$^-$ cells were sorted and RNA was extracted with a RNeasy Plus micro kit (Qiagen), then was reverse-transcribed with a high capacity cDNA reverse transcription kit (Applied Biosystems) and analyzed by quantitative RT-PCR with a Vii 7 Real-time PCR system (Applied Biosystems) with the following primers and probes (from Applied Biosystems; identifier in parentheses): *Ascl2* (Mm01268891_g1) *Ifng* (Mm00801778_m1), *Il17a*

(Mm00439619_m1), *Il21* (Mm00517640_m1), *Il4* (Mm00445259_m1), *Il21r* (Mm00600317_m1), *Il6* (Mm00446191_m1), *Il10* (Mm00439616_m1), *Wnt1* (Mm01300555_g1), *Wnt2* (Mm00470018_m1), *Wnt3a* (Mm00437337_m1), *Wnt4* (Mm01194003_m1), *Wnt5a* (Mm00437347_m1), *Wnt6* (Mm00437353_m1), *Wnt7a* (Mm00437356_m1), *Wnt7b* (Mm01301717_m1), *Wnt8a* (Mm01157914_g1), *Wnt8b* (Mm00442108_g1), *Wnt11* (Mm00437328_m1). The comparative threshold cycle method and the internal control *Gapdh* (Mm99999915_g1) was used for normalization of the target genes.

**TCRγδ$^+$CXCR5$^+$ vs. TCRγδ$^+$CXCR5$^-$ cell expression analysis**. nCounter mouse mRNA Assay Kit (NanoString Technologies) were used to detect mRNA from sorted TCRγδ$^+$CXCR5$^+$ and TCRγδ$^+$CXCR5$^-$ following the manufacturer's protocol. Assay and spike-in controls were used for normalization based on identical amounts of input RNA (60 ng/sample). The correlation of gene expression among cell types were calculated and plotted as heatmap using the Multi Experimental Viewer (MeV) software. Differentially expressed genes were identified by Student's *t*-test and statically relevant results consisted in *p*-value < 0.05.

**Statistics**. GraphPad Prism 7.0 was used for statistical analysis (unpaired, two-tailed Student's *t*-test or one-way ANOVA, followed by Tukey multiple comparisons). Differences were considered statistically significant with a *p* value of less than 0.05.

**Data availability**. The datasets included in this study are available as Supplementary Data 1 and 2.

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

## Acknowledgements

This work was supported by the NIH (R01 AI43458 to H.L.W.). We thank Deneen Kozoriz for her excellent technical support in cell sorting.

## Author contributions

R.M.R. initiated the project, designed the experiments, carried out most of the experiments, and wrote the manuscript. A.L., S.R., C.K., N.S., T.G.M., S.L., B.A.D., and G.G. helped perform the experiments. G.B.M. helped develop the whole lymph node confocal microscopy and provided input for the manuscript. H.L.W. supervised the experiments and the manuscript.

## Additional information

**Competing interests:** The authors declare no competing interests.

