## [Peer Review File · Nature Communications]

Reviewers' comments:

Reviewer #1 (gd T cell biology, cancer immunity)(Remarks to the Author):

The manuscript by Rezende, Weiner and coworkers reports a subset of follicular CXCR5+ $\gamma\delta$ T cells that controls Tfh cell differentiation and thereby impacts humoral responses. Mechanistically they suggest that the novel $\gamma\delta$ T cell subset produces Wnt8a/b that induces the expression of Ascl2 in Tfh cell precursors. Surprisingly, the CXCR5+ $\gamma\delta$ T cell subset can also act as an antigen-presenting cell, a rare property (seemingly restricted to the minor CXCR5+ subset) of murine $\gamma\delta$ T cells. The authors assess the production of allo and autoantibodies in TCR δ KO mice and their phenotype in an experimental model (pristine-induced) of lupus to attest the (patho)physiological relevance of their findings.

The paper provides novel insight into the previously reported but poorly understood (from a mechanistic point of view) role of $\gamma\delta$ T cells in shaping antibody responses. There are, however, four important issues that require additional clarification:

1. The differential involvement of gd T cells in B cell antibody production in CFA-induced versus Alum-induced responses. The explanations found in the Discussion, based on Alum being a weaker adjuvant or eliciting different cytokines, are not satisfactory and require mechanistic investigation regarding the specific link to gd T cells.
2. The differential involvement of gd T cells in class switching: why in IgG1 and IgE but not IgG3 or sIgA??
3. How physiologically relevant is the antigen presentation by gd T cells in vivo? What happens to the GC reaction when you transfer MHCII-/- gd T cells into TCRd-/- mice?
4. How physiologically relevant is CXCR5 expression by gd T cells in vivo? What happens to the GC reaction when you transfer CXCR5-/- gd T cells into TCRd-/- mice?

Reviewer #2 (Th differentiation)(Remarks to the Author):

This paper presents findings on a novel and significant immunological pathway by which gamma-delta (GD) T cells augment the Tfh/GC/Ab response. There have been hints of a role for GD T cells in Ab responses in earlier studies, but this manuscript fleshes out the whole pathway. The most novel and important findings in this study are that 1) GD T cells are essential for normal Tfh/GC/Ab responses when CFA is used as an adjuvant, 2) CXCR5+ GD T cells can present Ag to CD4 T cells, 3) GD T cells secrete Wnt ligands that can help promote Tfh cell differentiation. Overall, the results portray a consistent story and there is a good breadth in the types of analyses and extensive experiments that buttress the conclusions. That being said, some of the results seem not as robust as they could be, for instance some of the differences are often small with only 3-4 replicates. Some of these experiments are said to be repeated but it would be good to see the repeats particularly or the experiments in Figure 10.

Likewise, the glomerulus diameter data in Fig 2d is not very convincing.

The CXCR5 staining in Fig 9e is not very good. Perhaps CXCR5 mRNA could be shown as well or instead?

A key question for the results in Fig 10b is if GC B cells and Ab levels are increased along with Tfh.

Some of the data in the main figures could be moved to supplemental data, for instance Fig 3 is mostly negative data, and Fig 5 is not surprising given other data, and could be moved to suppl. Fig 6 is not very impressive and would be better in suppl. data. On the other hand, the data showing APC function for GD T cells in Suppl. fig 5 is important and would be better appreciated as a main figure.

Minor points:

Fig 9d-- why didn't DCs induce Cxcr5 in CD4 T cells? Or was that not done? If not done, why not?

Fig 4e-- what is the gray fluor used in the microscopy?

Fig 2a-c legend-- give some description of assay used to generate data

Introduction line 35-- IL-2 is mentioned in a way that makes it sound like IL-2 promotes Tfh cell differentiation but IL-2 inhibits Tfh cell differentiation.

Results line 235-237 -- somewhat confusing the way the pristane model is mentioned here. Clarify that this is a different experiment.

Reviewer #3 (gd T function, B/T interaction)(Remarks to the Author):

The manuscript by Rezende et al. addresses the important and timely question of the role of gammadelta (gd) T cells in the humoral immune responses. It describes an experimental tour de force ranging from basic studies of the effect of gd T cell deficiency on antibody levels (immunization-induced and autoantibodies) and the B cell compartment to cutting edge experiments examining the effect of gd T cells on TFH induction and B cell help, APC function of an inducible subset of gd T cells, and finally the molecular mechanism by which these gd T cell APCs drive TFH differentiation and function, through the secretion of Wnt ligands. As a "package", this study is very compelling, and its findings represent a major advance in the field. Overall, the experiments are well designed and presented, and interpretations seem reasonable and justified.

Details:

1) The manuscript in its current form is very long. This is partly due to the description of experiments that essentially repeat and confirm findings already published. For example that TCR-d-/- mice have a mostly normal B cell compartment in bone marrow and periphery, was reported in great detail by Huang et al., *J I* 196:217,2016). Of course the point is critical for the particular role assigned to gd T cells in this paper, but it could be made more briefly by referencing the Huang paper and, if deemed necessary, including confirmatory experiments with supplemental data. More generally, the paper should be trimmed wherever possible.

2) Because their publication will be seen by and should be useful to a broader non-expert readership, the authors ought to reveal "dirty little secrets". Specifically, it seems important to point out that CFA preferentially activates a particular subset of murine gd T cells, those expressing Vg4 (Roark et al., *J I* 179:5576, 2007), and that these same cells, through activation with CFA, are induced to express MHCII, and to function as APCs, as reported by Cheng et al., *J. Neuroimmunol.* 203:3, 2008. To my knowledge, this remains to be shown for other murine gd T cells. Furthermore, when induced by immunization with OVA/alum, cells of the same Vg4+ subset become suppressive for type 2 humoral immune responses (Huang et al., *J I* 183:849, 2009; Huang et al., *J I* 190:913, 2013). Therefore, the current report actually describes those humoral responses that are assisted by these particular gd T cells. This should be made very clear. Moreover, because Vg1+ gd T cells are known to enhance type 2 humoral responses, there is the intriguing possibility that distinct subsets of gd T cells, in combination with suitable adjuvants, help to determine the flavor of the humoral responses.

3) Some minor inaccuracies should be addressed: Reference #9 of the current manuscript does not

show reduced IgA levels in TCR-d^{-/-} mice (although certain other Igs are reduced), nor does it demonstrate that increased IgE levels in Vg4/6^{-/-} mice depend on IL-4 secretion by Vg1⁺ cells. Instead, it shows that IL-4 production by T cells (gd and ab) and total serum IL-4 are increased in Vg4/6^{-/-} mice.

Contrary to broad statements in the introduction, Huang et al. JI 183:849, 2009 already reported decreased levels of OVA-induced antibodies in TCR-d^{-/-} mice (Fig.1a), albeit using Alum as adjuvant.

The notion that all influence of gd T cells on antibody production goes through ab T cells is in contradiction to reported findings that total Ig levels as well as IgM, and possibly IgG2b/c change in mice with altered gd T cell composition, even when ab T cells are absent (Huang et al., PNAS 112, E39,2015).

The reference to HSP-65-reactive gd T cells in the discussion (#43 in the current manuscript) is misleading in the context of the CFA-dependent responses studied here because O'Brien et al. mainly found Vg1⁺ and never Vg4⁺ TCRs associated with this response. However, it might be relevant with pristane-induced reactivity?

It is also not entirely correct to assert that previously "the role of gd T cells in the humoral immune response has been primarily characterized in the context of global antibody production" (first paragraph of the results section). In fact, numerous early studies tested the idea that gd T cells can provide specific B cell help, with variable success. The novelty of the current paper does not stem from a demonstration that gd T cells can influence specific antibody production but rather from its characterization of the interaction between gd T cells, TFH cells and TFH-dependent humoral responses in the specific settings of CFA and Pristane-assisted immunity/autoimmunity.

Reviewer #1:

The manuscript by Rezende, Weiner and coworkers reports a subset of follicular CXCR5+ $\gamma\delta$ T cells that controls Tfh cell differentiation and thereby impacts humoral responses. Mechanistically they suggest that the novel $\gamma\delta$ T cell subset produces Wnt8a/b that induces the expression of Ascl2 in Tfh cell precursors. Surprisingly, the CXCR5+ $\gamma\delta$ T cell subset can also act as an antigen-presenting cell, a rare property (seemingly restricted to the minor CXCR5+ subset) of murine $\gamma\delta$ T cells. The authors assess the production of allo and autoantibodies in TCR δ KO mice and their phenotype in an experimental model (pristane-induced) of lupus to attest the (patho)physiological relevance of their findings.

The paper provides novel insight into the previously reported but poorly understood (from a mechanistic point of view) role of $\gamma\delta$ T cells in shaping antibody responses. There are, however, four important issues that require additional clarification:

1. The differential involvement of $\gamma\delta$ T cells in B cell antibody production in CFA-induced versus Alum-induced responses. The explanations found in the Discussion, based on Alum being a weaker adjuvant or eliciting different cytokines, are not satisfactory and require mechanistic investigation regarding the specific link to $\gamma\delta$ T cells.

Response: To address the reviewer's comment, we performed an additional series of experiments which establishes that Alum does not activate $\gamma\delta$ T cells and that non-activated $\gamma\delta$ T cells are dispensable for control of the humoral immune response.

Specifically:

(1) We show that CFA, but not Alum induced CXCR5 (**new Fig. 5a, b; new Supplementary Fig. 5b; new Supplementary Fig. 7c; new Supplementary Fig. 9a**) and MHC-II (**new Supplementary Fig. 9a**) on $\gamma\delta$ T cells. We also now demonstrate that both CXCR5 and MHC-II expression are critical for the ability of $\gamma\delta$ T cells to induce Tfh cell differentiation and germinal center formation (**new Fig. 4c, d; new Fig. 6f, g**);

(2) It has been shown that lower antibody levels in TCR δ ^{-/-} mice relates to the absence of the V γ 1 subset, which favors humoral immune responses, as opposed to the V γ 4 subset that suppresses humoral responses^{1, 2, 3, 4}. Thus, the balance between V γ 1 and V γ 4 $\gamma\delta$ T cell subset functions determines the level of antibody production following CFA vs. Alum immunization and could explain why $\gamma\delta$ T cells do not participate in antibody production in Alum immunized mice (Fig. 1). To test this, we depleted V γ 4 $\gamma\delta$ T cells with an anti-V γ 4-depleting mAb and found marked increase in all OVA-specific immunoglobulin subclasses following CFA immunization, but not following Alum immunization (**new Supplementary Fig. 1**). Thus, even in the absence of the inhibitory effects of V γ 4 $\gamma\delta$ T cells, the V γ 1 subset does not promote antibody production following Alum immunization. Moreover, Tfh cell frequency was higher in V γ 4 $\gamma\delta$ T cell-depleted mice than in isotype control-treated mice immunized with CFA/OVA, but not in mice immunized with Alum/OVA (**new Supplementary Fig. 7c**).

(3) Taken together, these data clearly demonstrate that Alum **does not** activate $\gamma\delta$ T cells and only activated $\gamma\delta$ T cells present antigens to naïve CD4 T cells and initiate the Tfh cell differentiation program.

2. The differential involvement of $\gamma\delta$ T cells in class switching: why in IgG1 and IgE but not IgG3 or sIgA? Response: To address this question as related to IgG3, we carried out additional experiments in which the number of mice was increased. We found that IgG3 levels were decreased in TCR δ ^{-/-} mice (**new Fig. 1g**). Regarding sIgA, the sIgA levels we reported in the previous version of the manuscript was total fecal IgA. Thus, we carried out a new experiment in which we measured sIgA levels in feces from TCR δ ^{-/-} mice using a ELISA kit specific for sIgA and found decreased sIgA levels in TCR δ ^{-/-} mice (**new Fig. 1d**).

3. How physiologically relevant is the antigen presentation by $\gamma\delta$ T cells in vivo? What happens to the GC reaction when you transfer MHCII^{-/-} $\gamma\delta$ T cells into TCR δ ^{-/-} mice? Response: To answer this question, as suggested by the reviewer, we transferred total $\gamma\delta$ T cells from MHC-II^{-/-} mice into TCR δ ^{-/-} mice and immunized with CFA. We found that TCR δ ^{-/-} mice transferred with $\gamma\delta$ T cells from MHC-II^{-/-} mice had reduced GC formation and decreased Tfh cell frequency as compared to TCR δ ^{-/-} mice that received $\gamma\delta$ T cells from WT mice (**new Fig. 6f, g**). This demonstrates that the APC function of $\gamma\delta$ T cells is critical for Tfh cell differentiation and GC formation.

4. How physiologically relevant is CXCR5 expression by $\gamma\delta$ T cells in vivo? What happens to the GC reaction when you transfer CXCR5^{-/-} $\gamma\delta$ T cells into TCR δ ^{-/-} mice? Response: To address this question, we transferred total $\gamma\delta$ T cells from CXCR5^{-/-} mice into TCR δ ^{-/-} mice and immunized with CFA, We found that TCR δ ^{-/-} mice transferred with $\gamma\delta$ T cells from CXCR5^{-/-} mice had reduced GC formation and decreased Tfh cell frequency as compared to TCR δ ^{-/-} mice that received $\gamma\delta$ T cells from WT mice. (**new Fig. 4c, d**) Thus, CXCR5 expression on $\gamma\delta$ T cells is crucial for $\gamma\delta$ T cell ability to induce Tfh cell differentiation and GC formation.

Reviewer #2:

This paper presents findings on a novel and significant immunological pathway by which gamma-delta (GD) T cells augment the Tfh/GC/Ab response. There have been hints of a role for GD T cells in Ab responses in earlier studies, but this manuscript fleshes out the whole pathway. The most novel and important findings in this study are that 1) GD T cells are essential for normal Tfh/GC/Ab responses when CFA is used as an adjuvant, 2) CXCR5⁺ GD T cells can present Ag to CD4 T cells, 3) GD T cells secrete Wnt ligands that can help promote Tfh cell differentiation. Overall, the results portray a consistent story and there is a good breadth in the types of analyses and extensive experiments that buttress the conclusions.

1. That being said, some of the results seem not as robust as they could be, for instance some of the differences are often small with only 3-4 replicates. Some of these experiments are said to be repeated but it would be good to see the repeats particularly for the experiments in Figure 10. Response: Regarding

figure 10 (**new Figure 8**), rather than showing a representative figure, we combined data from 2-3 independent experiments that strengthen the results.

2. Likewise, the glomerulus diameter data in Fig 2d is not very convincing.

Response: All kidney histologic slides show consistent diminished glomerulus diameter and cellularity in TCR δ ^{-/-} mice upon pristane treatment. We present a representative image in Fig. 2d.

3. The CXCR5 staining in Fig 9e is not very good. Perhaps CXCR5 mRNA could be shown as well or instead?

Response: Fig. 9d (**now new Fig. 7d**) shows CXCR5 mRNA levels in CD4 T cells co-cultured in the same conditions as in Fig. 9e (**new Fig. 7e**), except for anti-Wnt8b treatment, since we did not detect CXCR5 mRNA in CD4 T cells co-cultured in the presence of anti-Wnt8 mAb.

4. A key question for the results in Fig 10b is if GC B cells and Ab levels are increased along with Tfh.

Response: We repeated the experiment shown in Fig. 10b (**now Fig. 8b**) and investigated OVA-specific antibody levels 21 days post CFA/OVA immunization. Germinal center staining was not performed because germinal center reactions are over at 21 days post immunization (when the experiment ended). Measurement of antibody levels at this time provides evidence of a previous germinal center reaction. We found that TCR δ ^{-/-} mice transferred with $\gamma\delta$ T cells from C59-treated WT mice had diminished anti-OVA total IgG and IgM levels as compared to TCR δ ^{-/-} mice that received cells from vehicle-treated WT mice. This indicates that blocking Wnt ligand secretion from $\gamma\delta$ T cells affects Tfh cell differentiation, which results in a defective germinal center formation and reduced serum antibody levels.

5. Some of the data in the main figures could be moved to supplemental data, for instance Fig 3 is mostly negative data, and Fig 5 is not surprising given other data, and could moved to suppl. Fig 6 is not very impressive and would be better in suppl. data. On the other hand, the data showing APC function for GD T cells in Suppl. fig 5 is important and would be better appreciated as a main figure.

Response: We agree with the reviewer and Figures 3 and 5 have been moved to supplementary data (**new Supplementary Fig. 2 and 6**). We moved Fig. 6c-e to Supplementary Fig. 7d-f and added new data on previous Fig. 6 (**new Fig. 4**). $\gamma\delta$ T cell APC-related functions from Supplementary Fig. 5 were moved to the main figures (**new Fig. 6c,d**).

Minor points:

1. Fig 9d-- why didn't DCs induce Cxcr5 in CD4 T cells? Or was that not done? If not done, why not?

Response: Because CXCR5 mRNA expression levels in CD4 T cells co-cultured in the presence of DCs were low, the graph scale we used precluded putting the results on the graph. We have adjusted the graph scale and now demonstrate that CD4 T cells in the presence of DCs do not upregulate CXCR5. We believe this is because DCs do not secrete Wnt8, which is crucial to initiate Tfh cell differentiation. DCs are important to induce the transcription factor Bcl6 on CXCR5-expressing CD4 T cells in order to complete the Tfh differentiation program.

2. Fig 4e-- what is the gray fluor used in the microscopy? Response: CXCR5-PE.

3. Fig 2a-c legend-- give some description of assay used to generate data. Response: We have added a description of the assay used to generate data in the Figure legend of **new Fig. 2a-c**.

4. Introduction line 35-- IL-2 is mentioned in a way that makes it sound like IL-2 promotes Tfh cell differentiation but IL-2 inhibits Tfh cell differentiation. Response: The reviewer is correct. We have clarified this in the revised manuscript.

5. Results line 235-237 -- somewhat confusing the way the pristane model is mentioned here. Clarify that this is a different experiment. Response: We have clarified this in the revised manuscript.

Reviewer #3:

The manuscript by Rezende et al. addresses the important and timely question of the role of gammadelta (gd) T cells in the humoral immune responses. It describes an experimental tour de force ranging from basic studies of the effect of gd T cell deficiency on antibody levels (immunization-induced and autoantibodies) and the B cell compartment to cutting edge experiments examining the effect of gd T cells on TFH induction and B cell help, APC function of an inducible subset of gd T cells, and finally the molecular mechanism by which these gd T cell APCs drive TFH differentiation and function, through the secretion of Wnt ligands. As a "package", this study is very compelling, and its findings represent a major advance in the field. Overall, the experiments are well designed and presented, and interpretations seem reasonable and justified.

Details:

1. The manuscript in its current form is very long. This is partly due to the description of experiments that essentially repeat and confirm findings already published. For example that TCR-d-/- mice have a mostly normal B cell compartment in bone marrow and periphery, was reported in great detail by Huang et al., JI 196:217,2016). Of course the point is critical for the particular role assigned to gd T cells in this paper, but it could be made more briefly by referencing the Huang paper and, if deemed necessary, including confirmatory experiments with supplemental data. More generally, the paper should be trimmed wherever possible. Response: We thank the reviewer for the comment and we have shortened the revised manuscript whenever possible.

2. Because their publication will be seen by and should be useful to a broader non-expert readership, the authors ought to reveal "dirty little secrets". Specifically, it is seems important to point out that CFA preferentially activates a particular subset of murine gd T cells, those expressing Vg4 (Roark et al., JI 179:5576, 2007), and that these same cells, through activation with CFA, are induced to express MHCII, and to function as APCs, as reported by Cheng et al., J.Neuroimmunol.203:3, 2008. To my knowledge, this remains to be shown for other murine gd T cells. Furthermore, when induced by immunization with OVA/alum, cells of the same Vg4+ subset become suppressive for type 2

humoral immune responses (Huang et al., JI 183:849, 2009; Huang et al., JI 190:913, 2013). Therefore, the current report actually describes those humoral responses that are assisted by these particular gd T cells. This should be made very clear. Moreover, because Vg1+ gd T cells are known to enhance type 2 humoral responses, there is the intriguing possibility that distinct subsets of gd T cells, in combination with suitable adjuvants, help to determine the flavor of the humoral responses. Response: We thank the reviewer for these comments. To better understand the involvement of $\gamma\delta$ T cell subtypes in the humoral immune response, we immunized WT mice with either CFA or Alum and investigated MHC-II and CXCR5 expression on V γ 1 and V γ 4 subsets. We found that only V γ 1+ $\gamma\delta$ T cells upregulated MHC-II expression upon CFA, but not Alum immunization (MHC-II^{high}; **new Supplementary Fig. 9a**). Importantly, CXCR5 was only expressed on MHC-II^{high}V γ 1+ cells (new Fig. 4b and new Supplementary Fig. 9a). Furthermore, depletion of V γ 4 $\gamma\delta$ T cells by an anti-V γ 4 depleting mAb markedly increased OVA-specific immunoglobulin following CFA, but not Alum immunization. This suggests that the V γ 4 $\gamma\delta$ T cell subtype has an inhibitory effect on V γ 1-induced antibody production. Our finding that CFA induces MHC-II on V γ 1 cells is different from the results of Roark et al., 2007. We believe this difference may be related to the different mouse strain used in that study (DBA/1 lac J) vs. the strain we used in our study (C57BL/6J). However, it is also possible that the APC functions of $\gamma\delta$ T cells upon CFA immunization described by Cheng et al., 2008 could be related to V γ 1 cells, since the differential role of $\gamma\delta$ T cell subtypes was not explored in that study. Thus, we conclude that V γ 1 $\gamma\delta$ T cells stimulate antibody production by **both** inducing Tfh cell differentiation, as shown in the present study, and by secreting cytokines that favor germinal center formation (demonstrated by Willi Born's and Rebecca O'Brien's group), whereas V γ 4 $\gamma\delta$ T cells suppress V γ 1 activity. We agree that depending on the mouse strain, adjuvant used for immunization and the disease model, distinct subsets of $\gamma\delta$ T cells determine the "flavor" of the humoral immune response.

3. Some minor inaccuracies should be addressed:

a) Reference #9 of the current manuscript does not show reduced IgA levels in TCR-d/- mice (although certain other Igs are reduced), nor does it demonstrate that increased IgE levels in Vg4/6/- mice depend on IL-4 secretion by Vg1+ cells. Instead, it shows that IL-4 production by T cells (gd and ab) and total serum IL-4 are increased in Vg4/6/- mice. Response: The reviewer is correct and we have now accurately described reference #9 in the revised manuscript.

b) Contrary to broad statements in the introduction, Huang et al. JI 183:849, 2009 already reported decreased levels of OVA-induced antibodies in TCR-d/- mice (Fig.1a), albeit using Alum as adjuvant. The notion that all influence of gd T cells on antibody production goes through ab T cells is in contradiction to reported findings that total Ig levels as well as IgM, and possibly IgG2b/c change in mice with altered gd T cell composition, even when ab T cells are absent (Huang et al., PNAS 112, E39,2015). Response: We agreed with the reviewer and we have addressed these differences in the Introduction and Discussion of the revised manuscript.

c) The reference to HSP-65-reactive gd T cells in the discussion (#43 in the current manuscript) is misleading in the context of the CFA-dependent responses studied here because O'Brien et al. mainly found Vg1+ and never Vg4+ TCRs associated with this response. However, it might be relevant with pristane-induced reactivity? Response: Because we showed the important role of V γ 1 $\gamma\delta$ T cells in mediating antibody production and because CFA induced both MHC-II and CXCR5 on V γ 1 $\gamma\delta$ T cells, we believe that the Hsp65 component of the MT present in the CFA is likely to be responsible for these effects. This is consistent with the work by O'Brien et al. (ref #45) describing V γ 1 as the $\gamma\delta$ T cells responsive to Hsp60.

d) It is also not entirely correct to assert that previously "the role of gd T cells in the humoral immune response has been primarily characterized in the context of global antibody production" (first paragraph of the results section). In fact, numerous early studies tested the idea that gd T cells can provide specific B cell help, with variable success. The novelty of the current paper does not stem from a demonstration that gd T cells can influence specific antibody production but rather from its characterization of the interaction between gd T cells, TFH cells and TFH-dependent humoral responses in the specific settings of CFA and Pristane-assisted immunity/autoimmunity. Response: The reviewer is correct. We have removed this sentence from the revised manuscript.

References

1. Huang, Y. *et al.* The influence of IgE-enhancing and IgE-suppressive gammadelta T cells changes with exposure to inhaled ovalbumin. *J Immunol* **183**, 849-855 (2009).
2. Huang, Y. *et al.* gammadelta T cells affect IL-4 production and B-cell tolerance. *Proc Natl Acad Sci U S A* **112**, E39-48 (2015).
3. Huang, Y. *et al.* Antigen-specific regulation of IgE antibodies by non-antigen-specific gammadelta T cells. *J Immunol* **190**, 913-921 (2013).
4. Hahn, Y.S. *et al.* Different potentials of gamma delta T cell subsets in regulating airway responsiveness: V gamma 1+ cells, but not V gamma 4+ cells, promote airway hyperreactivity, Th2 cytokines, and airway inflammation. *J Immunol* **172**, 2894-2902 (2004).

REVIEWERS' COMMENTS:

Reviewer #1 (Remarks to the Author):

The authors addressed all my concerns by performing new experiments and adding the interesting data to the revised manuscript, which is now significantly improved and suitable for publication.

Reviewer #2 (Remarks to the Author):

This revised manuscript presents findings on a novel and significant immunological pathway by which gamma-delta (GD) T cells augment the Tfh/GC/Ab response. There have been hints of a role for GD T cells in Ab responses in earlier studies, but this manuscript dissects out a whole novel pathway. The most novel and important findings in this study are that

- 1) GD T cells are essential for normal Tfh/GC/Ab responses when CFA is used as an adjuvant,
- 2) CXCR5+ GD T cells can present Ag to CD4 T cells via class II MHC,
- 3) GD T cells secrete Wnt ligands that can help promote Tfh cell differentiation.

Overall, the results are coherent and there is strong breadth in the types of analyses and experiments that support the conclusions. The revisions have improved the manuscript substantially and the reviewers have addressed all my concerns in a satisfactory way.

Reviewer #3 (Remarks to the Author):

Fine study. No additional comments.